# Novel methods to study sea ice deformation, Linear Kinematic Features and Coherent Dynamic Clusters from imaging remote sensing data

Polona Itkin[1,2]

[1]UiT The Arctic University of Norway, Tromsø, Norway
[2]Norwegian Polar Institute, Tromsø, Norway

**Correspondence:** Polona Itkin (polona.itkin@npolar.no)

**Abstract.** Satellite Synthetic Aperture Radar (SAR) data are commonly utilized for calculating sea ice displacements and, consequently, sea ice deformation strain rates. However, strain rate calculations often suffer from a poor signal-to-noise ratio, especially for products with a spatial resolution higher than 1 km. In this study, a new filtering method to strain rate calculations derived from Sentinel-1 SAR image pairs with a spatial resolution of 800 m was applied. Subsequently, a power law to evaluate the deformation rates at decreasing spatial resolutions was employed to assess the quality of the filtered data. Upon positive evaluation of the filtered data, two innovative methods for sea ice deformation assessment were introduced. The first method, named 'damage parcels' tracking, involved the combined analysis of displacements and deformation strain rates to monitor divergence and convergence within the sea ice cover. Additionally, a new term to describe the behavior of the winter pack was proposed: 'Coherent Dynamic Clusters' (CDC). CDCs are cohesive clusters of ice plates within the pack ice that move coherently along Linear Kinematic Features (LKFs). The second novel method developed in this study focused on exploring the geometrical properties of these CDCs. Both methods were applied to the January-February collection of Sentinel-1 SAR imagery available during the N-ICE2015 campaign. The damage parcels were continuously tracked over a period of three weeks, including a major storm, revealing a slow healing process of existing LKFs. Furthermore, the CDC analysis demonstrated the presence of elongated CDCs with a density ranging from 5 to 20 per 100 km by 100 km, and the shortest distance between LKFs was found to be 5-10 km.

*Copyright statement.* TEXT

## 1 Introduction

Sea ice deformation has wide-ranging implications for climate, biology, and navigation. For example, occurrence of leads locally increases the thermodynamic coupling between the atmosphere and ocean, increases light availability for the primary production and aids the navigation. On contrary, does the occurrence of pressure ridges increase the surface roughness and dynamical coupling between the atmosphere and ocean, provide protective habitat for the life in the ocean, and obstruct the

navigation. With the Arctic ice thinning, becoming more seasonal, and increasing in mobility (Meredith and Schuur, 2019), the processes of sea ice deformation are likely intensifying (Rampal et al., 2009; Olason and Notz, 2014; Itkin et al., 2017). Deformation changes sea ice thickness instantly (Kwok and Cunningham, 2015; Itkin et al., 2018; von Albedyll et al., 2022),

and gradually, though preferential melt rates of pressure ridges (e.g. Salganik et al., 2023b) , both influencing the state of sea ice, critical for accurate sea ice forecasting and projections (Bushuk et al., 2017; Tian et al., 2021). To monitor sea ice states and developments, numerical models and satellite remote sensing products are indispensable tools. However, existing sea ice rheologies in numerical models, such as those by Hibler (1979) , Hunke and Dukowicz (1997), Heorton et al. (2018) and Ólason et al. (2022), do not entirely align with satellite-derived kinematic properties (Hutter et al., 2022). Furthermore, reliable

detection of deformed ice and leads (Zakhvatkina et al., 2019; Lohse et al., 2020; Guo et al., 2023), sea ice roughness (Farrell et al., 2020), and sea ice thickness (Ricker et al., 2017) through satellite remote sensing techniques remains challenging and is plagued by significant uncertainties (Zygmuntowska et al., 2014; Landy et al., 2020).

Decades of research on sea ice deformation have yielded significant progress, leading to a deep understanding of its multi-fractal nature, as extensively documented in the literature (Erlingsson, 1988; Schulson, 2004; Marsan et al., 2004; Hutchings

et al., 2011; Itkin et al., 2017; Oikkonen et al., 2017). This body of knowledge reveals that various features, such as small cracks approximately 1 m wide, ridges with a width of 10 m, leads spanning 100 m or more, and even complex systems of Linear Kinematic Features (LKFs) , first described by Kwok (2001), measuring 1 km or wider, exhibit similar characteristics in both space and time. This self-similarity extends not only to their shapes but also to the strain rates within these fractures. Such patterns can be described using scaling laws, enabling the measurement of deformation at a specific spatial or temporal resolution

and facilitating comparisons between different datasets, regions, and seasons. These comparisons, in turn, are instrumental for evaluating numerical models (Rampal et al., 2019; Hutter et al., 2022). Furthermore, it has been observed that sea ice fractures of various spatial scales tend to occur at typical intersection angles (Erlingsson, 1988; Hutter et al., 2022; Ringeisen et al., 2023). These fractures align along the Mohr-Coulomb failure lines, dividing the ice surface into distinct, parallelogram-shaped plates that move relative to one another, much like tectonic plates on a planet. (Erlingsson, 1988; Schulson, 2004; Dansereau

et al., 2019). In this paper the cohesive clusters of these plates are called 'Coherent Dynamic Clusters' (CDC) - a name that describes the transient nature of their motion along the fractures. The CDCs can be described by size and shape parameters, offering a novel option for sea ice deformation characterization. Motion of CDCs along the fractures can persist for several days (Coon et al., 2007; Graham et al., 2019), although strain rates typically cycle between divergence and convergence over the course of a weather event (Graham et al., 2019). The collision and sliding along these fractures cause damage to the CDC

edges, manifesting as leads and ridges. Over time, these features can consolidate by healing at low temperatures (Oikkonen et al., 2016) or through the injection of freshwater from snowmelt in the spring (Salganik et al., 2023a). As a result, the consolidated ridges and refrozen leads in CDCs reflect past deformation events. At any given moment, these CDCs may separate along newly formed fractures in either undamaged or previously damaged ice. Ridges and leads can also reactivate (Oikkonen et al., 2017), as any ice damage presents a weak point in the ice cover that can persist over seasonal timescales. Thus, damage should

be considered an additional parameter - alongside thickness and roughness - emerging from sea ice deformation. Research

has shown that incorporating damage into numerical models of sea ice (Girard et al., 2011) can enhance the representation of deformation and improve sea ice thickness distribution in modeling efforts (e.g. Ólason et al., 2022).

Although deformation can be estimated from sequential buoy positions (Rampal et al., 2009; Hutchings et al., 2011; Itkin et al., 2017) or obtaining displacements by comparing images, such as those from X-band ship radar (Oikkonen et al., 2017), much of the progress in this field is reliant on a vast volume of C-band satellite Synthetic Aperture Radar (SAR) data, including data from RADARSAT program of the Canadian Space Agency and Sentinel-1 mission of the European Space Agency. Sea ice displacements can be determined from sequential SAR images using feature or pattern tracking algorithms (Hollands and Dierking, 2011; Komarov and Barber, 2014; Korosov and Rampal, 2017), which are then utilized to calculate strain rates (Kwok, 1998; Kwok and Cunningham, 2015; von Albedyll et al., 2021). Specifically, the RADARSAT Geophysical Processor System (RGPS) (Kwok, 1998) has been instrumental in deriving properties such as scaling laws in spatio-temporal distribution (Marsan et al., 2004; Bouillon and Rampal, 2015; Rampal et al., 2019) and patterns in LKFs (Hutter et al., 2022). However, the spatial resolution of deformation estimated from SAR has been constrained to coarser than 1 km by signal-to-noise ratio (von Albedyll et al., 2021; Ringeisen et al., 2023).

The aim of this paper was to explore the potential use of Sentinel-1 SAR data at higher spatial resolutions to investigate sea ice deformation properties. This paper utilized sea ice deformation data and findings from the N-ICE2015 expedition conducted in January and February 2015 in the pack ice north of Svalbard, as reported by Granskog et al. (2018), Itkin et al. (2017), Oikkonen et al. (2017) and Graham et al. (2019). The methodology involved comparing the power law of SAR-derived strain rates with other N-ICE2015 data. In the next phase, this study examined if SAR data can be employed to track damage along LKFs between temporally separate weather events. Finally, the paper explored the possibility of detecting CDCs.

## 2 Data

The N-ICE2015 expedition (Granskog et al., 2018), conducted aboard RV Lance as it drifted freely with the pack ice, provided an exceptional dataset on sea ice deformation. This expedition marked a historic moment as it enabled simultaneous high-resolution deformation observations from GPS buoys (Itkin et al., 2015), ship radar (Haapala et al., 2017), and satellite SAR. In this paper, data from the winter leg (leg 1) spanning from mid-January to mid-February was utilized. During this period, the pack ice in the N-ICE2015 region consisted mainly of second-year ice to the north and northwest of RV Lance, while first-year ice was prevalent in the south and southeast (Itkin et al., 2017). Over the course of the month analyzed in this study, the distance between the ship and the ice edge decreased from 200 km to 50 km. Additionally, several storms, which induced significant sea ice deformation, passed over the pack ice during this time frame (Graham et al., 2019). For a comprehensive understanding of sea ice deformation during N-ICE2015, readers are encouraged to refer to the works of Itkin et al. (2017) and Oikkonen et al. (2017). Further details regarding the expedition's atmosphere-ice-ocean interactions can be found in the summary provided by Graham et al. (2019).

Central to this paper are the 145 Sentinel-1A Synthetic Aperture Radar (SAR) images obtained from the CREODIAS 2.0 server (cre, 2023), tracking the same sea ice area during the N-ICE2015 expedition north of Svalbard from 15 January to 18

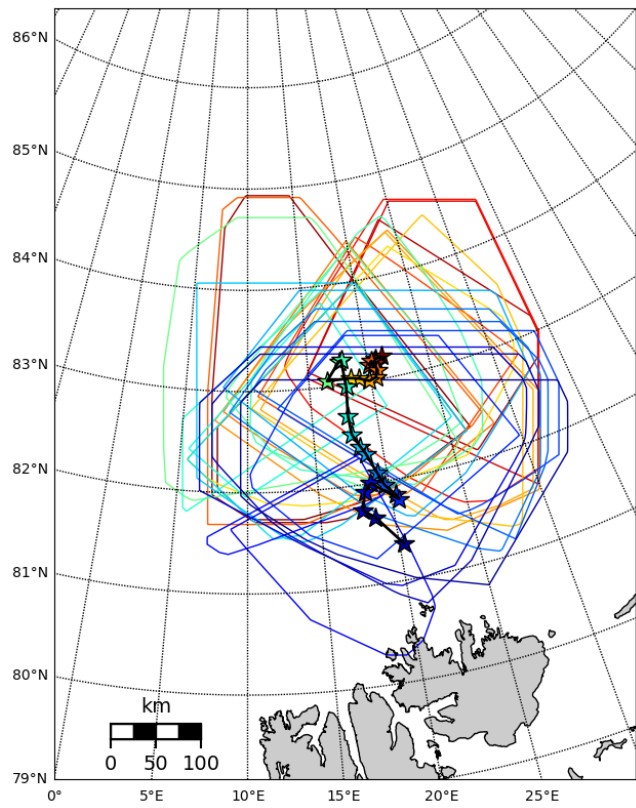

**Figure 1.** The N-ICE2015 RV Lance daily positions (stars) from 15 January to 18 February. The ship drifted southwards and colors of stars are increasing with time (jet colormap). The frames show total coverage of SAR sea ice deformation dataset for each day in the radius of 200km from RV Lance. By 27 February ship reached the ice edge and was relocated back towards 83°N,25°E.

February 2015. Additional 49 images were downloaded to extend the CDC analysis until 27 February. These images were
among the initial operational data from the Sentinel-1A mission. All the satellite images were acquired in Extra Wide Swath
mode, the typical mode activated over sea ice, providing Level-1 Ground Range Detected (GRD) images with a 410 km wide
swath and a pixel size of 40 m by 40 m. The advantageous high latitude location of the expedition allowed for data collection in
two temporal clusters of the polar-orbiting satellite: the ascending orbit in the early morning (5 to 8 UTC) and the descending
orbit in the afternoon (13 to 16 UTC). However, there were several episodes during the month when GRD data were unavailable,
notably between 27 January and 3 February.

The ship radar data from N-ICE2015 were obtained through the digitizing unit of the navigational X-band radar installed on
RV Lance Karvonen (2016); Haapala et al. (2017). The radar antenna, positioned at a considerable height on the ship's mast,
offered a range of 15 km with minimal mast shading and a spatial resolution of 12.5 m. For this study, the processed total
deformation rates derived from daily radar image pairs, as processed by Oikkonen et al. (2017), were utilized. To align with the
temporal coverage of the Sentinel-1 data, the week between 27 January and 3 February was excluded for comparative analysis.

During the N-ICE2015 expedition, an extensive array of GPS buoys was deployed in two concentric rings around RV Lance, transmitting data at hourly and 3-hourly intervals (Itkin et al., 2015). For this study, only the inner ring, with a diameter of approximately 20 km and comprising 11 buoys, was analyzed. The external ring of buoys was incomplete towards the north and thus excluded from the analysis.

## 3   Methods

All datasets were processed into spatially distributed points, capturing known displacements at defined, ideally daily, timesteps. The highest spatial scales of the ship radar and buoy data were determined based on signal-to-noise ratios, following the approach suggested by Hutchings et al. (2012). For more detailed information on the processing of ship radar and buoy data, readers can refer to Oikkonen et al. (2017) and Itkin et al. (2017).

Sea ice displacements from SAR image pairs were estimated using a sequence of feature tracking (FT) and pattern matching (PM) methods developed by Korosov and Rampal (2017). While not calibrated, each image was multi-looked, averaging radar intensity values over an 80 m by 80 m area. This process helped mitigate speckle noise that could introduce errors in the geographical positioning of features or patterns in the image. In the subsequent step, the FT method provided the initial estimate of displacements on an irregular grid for each SAR image pair. These initial estimates were used to narrow the local search radius, improving the computational efficiency of the subsequent PM method applied to the same image pair. For PM, a regular orthogonal grid with points separated by 800 m (10 pixels) was seeded. Using a regular grid was advantageous for sea ice deformation calculations as it provided the foundation for a triangulation mesh with elements of comparable length scales. In the PM method, image sub-samples (templates) measuring 3.2 km by 3.2 km (45 by 45 pixels) were matched based on mean displacements from FT. Maximum cross-correlation (MCC) with rotation was utilized to estimate displacements, and a threshold value of the Hessian matrix of the MCC was empirically determined and employed as a quality control measure. Displacements with values lower than this threshold were discarded, effectively removing geometrical artifacts such as image edges.

The estimation of sea ice deformation from sea ice displacements followed the same method for all three datasets. The displacements were converted into average velocities by dividing them by the time differences between the recorded positions. Initially, the origin displacement location points were triangulated using Delaunay triangulation, where triangles with one of the angles sharper than 15 degrees were discarded. Sea ice deformation was then estimated using the commonly-used line integrals of Green's theorem (Marsan et al., 2004; Hutchings and Hibler III, 2008). Shear and divergence were estimated from spatial derivatives of velocities: $u_x$, $u_y$, $v_x$ and $v_y$. $u_x$ is defined as:

$$u_x = \frac{1}{A} \Sigma_{i=1}^{n} (u_{i+1} + u_i)(y_{i+1} - y_i) \tag{1}$$

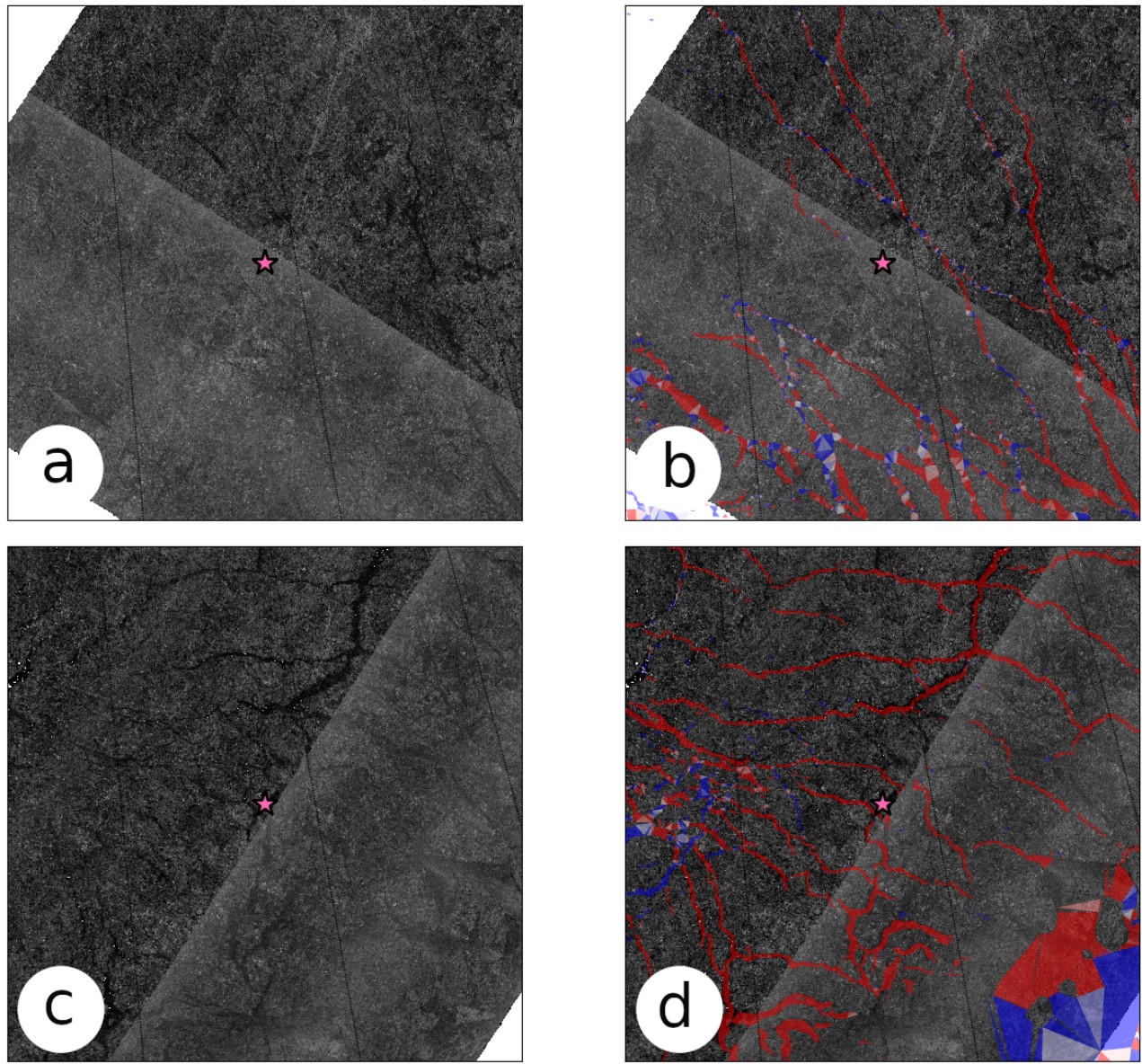

**Figure 2.** Second SAR image in the pair (a, c) overlaid by calculated deformation from the image pair (b, d) for 25 January (a, b) and 9 February 2015 (c, d). The sizes of the frames are 200km by 200 km. The location of RV Lance is represented by a pink star in the centers of the frames.

, where $A$ is is the area of triangle, $i$ is the index of a corner of triangle, $n = 3$ and $n + 1 = 1$. The other derivatives are defined in a similar way. Then the shear $\epsilon_{SHR}$, divergence $\epsilon_{DIV}$ and total deformation rate $\epsilon_{TOT}$ are defined as:

$$\epsilon_{SHR} = \sqrt{(u_x - v_y)^2 + (u_y + v_x)^2} \tag{2}$$

,

$$\epsilon_{DIV} = u_x + v_y \tag{3}$$

,

$$\epsilon_{TOT} = \sqrt{\epsilon_{DIV}^2 + \epsilon_{SHR}^2} \tag{4}$$

.

## 3.1 Additional SAR sea ice deformation processing

In this study, an 800 m distance between grid points was chosen as the highest resolution for the SAR data analysis. Although
a higher resolution was technically feasible, it would not be advantageous for the comparison with the ship radar and buoy
data. The 800 m grid spacing aligns with one of the shorter length scales of ship radar data and is wider than most of the leads
and Linear Kinematic Features (LKFs) observed during N-ICE2015, as reported by Graham et al. (2019). Opting for a shorter
distance would have limited the sampling of the shortest distance ($\lambda$) over these fracture zones, compromising the accuracy of
the power law scaling analysis.

The accuracy of the data, denoted as $\sigma_x$, can be estimated as 80 m, representing the lowest detectable displacement. Follow-
ing the noise-signal ratio formulation by Hutchings et al. (2012), sea ice deformation calculations become reasonably noise-free
when $A >> 8n^2\sigma_x^2$, where $A$ is the area of the polygon with n-number of nodes (3 nodes for a triangle). If $\lambda = \sqrt{A}$, this con-
dition is satisfied only when $\lambda >> 679$ m, for example at an order of magnitude larger values of 5-10 km. Consequently,
sea ice deformation calculations at shorter length scales are not reliable. This limitation is clearly visible in Figure **??** a. The
noise arises from undetectable displacements shorter than the pixel size ($\sigma_x$) of the SAR images. These short displacements
are extremely low or even close to zero. The residual deformation resulting from these displacements accumulates in artifacts
with rhomboid shapes. These shapes appear due to the step function in the sea ice displacement algorithm: the value of a dis-
placement can only increase by $\sigma_x$. These increases occur perpendicular and parallel to the template rotation in PM, creating
a rhomboid pattern that becomes denser in regions where the velocity gradients are larger. It's crucial to note that this pattern
should not be confused with similarly-shaped fractures following Mohr-Coulomb failure lines in Linear Kinematic Features
(LKFs) (Erlingsson, 1988; Schulson, 2004; Dansereau et al., 2019). Realistic features, however, stand out as locations with
a stronger signal, correctly spatially confined to leads (see Figure 2). Calculating deformation at a coarser spatial resolution,
which would smooth the rhomboid artifacts, would inevitably lead to the loss of information about the location of deforming
sea ice. As demonstrated by Korosov and Rampal (2017) using the example of coastline locations, the sea ice displacement
algorithm exhibited a spatial accuracy of approximately 200 m, significantly higher than the 800 m resolution used in this
study.

To examine if also the deformation values in those features were realistic they were isolated by applying a detection limit $DL$ defined as:

$$DL = nk\sigma_x^2/(2Adt), \tag{5}$$

where $k$ is the scaling coefficient of $\sigma_x$, and $dt$ is the time difference between the images in the image pair. The rhomboid artifacts have values right at the $DL$ for $k = 1$ and can be efficiently removed by threshold $DL$ for e.g. $k = 1.3$. In this study this increases the $\sigma_x$ by 30% from 80 m to approximately 120 m.

In addition to the noise-signal ratio issues in the displacement data, another source of error in the deformation estimates by line integrals stems from the orientation of the Linear Kinematic Features (LKFs) concerning the triangle boundaries. Equation 1 assumes a homogeneous velocity gradient along the boundary, a condition often violated in triangles - polygons with just three nodes, where an LKF can cross its boundary under any angle. This 'boundary-definition error' is a well-known problem (Lindsay and Stern, 2003; Bouillon and Rampal, 2015) that can lead to spurious opening and closing along the LKFs. While this problem can be mitigated by isotropic smoothing (Lindsay and Stern, 2003), Bouillon and Rampal (2015) suggested a directional filtering of deformation values of triangles specifically along the LKFs. Such anisotropic smoother, here called the LKF filter (LKFF) follows the direction of the LKF and preserves the accurate information of the deformation localization. LKFF was defined by the size of the kernel, the number of boundary crossed in each direction, and its minimal size. LKFF was applied to the data previously filtered by DL.The kernel size suggested by (Bouillon and Rampal, 2015) was 3, for a dataset with 10-km grid spacing. In this study, the spacing is much shorter (800 m) and the smallest possible kernel of 1 boundary crossing was used. For a single LKF this resulted in LKFF size of 3 triangles, while for a complex case of LKF crossing the kernel size may be larger. Only deformation features with at least one entire kernel size were considered, while the others were removed. This eliminated the rhomboid artifacts that persisted after the DL filtering, owing to accumulated noise in the 'corners' of the rhomboids, where two directions of the step function crossed. LKFF averages out the spurious switching between divergence and convergence from one triangle to the next (compare Figure 3 a and b) and reduces the deformation values. As triangles along the LKFs may have varying sizes, the averages were weighted by the triangle area.

The overview of the methods for SAR sea ice deformation used in this paper is given in Figure 3.

Each SAR image pair had a restricted spatial extent and occasionally did not cover the entire sea ice surface. To expand the data coverage, several surrounding image pairs were processed for deformation and then tiled together. Since the time difference between the pairs varied, different $DL$ values were correspondingly applied. Figures 2 and 3 already display such a tiled product.

To facilitate comparison, SAR-derived deformation values were coarse-grained from the original 800 m resolution into logarithmically spaced $\lambda$, similar to the other two data sources. The minimum spatial coverage of any coarse-level triangle by fine triangles was set at 10%, and the assigned values were means weighted by the spatial coverage. Triangles with spatial coverage below this threshold were discarded.

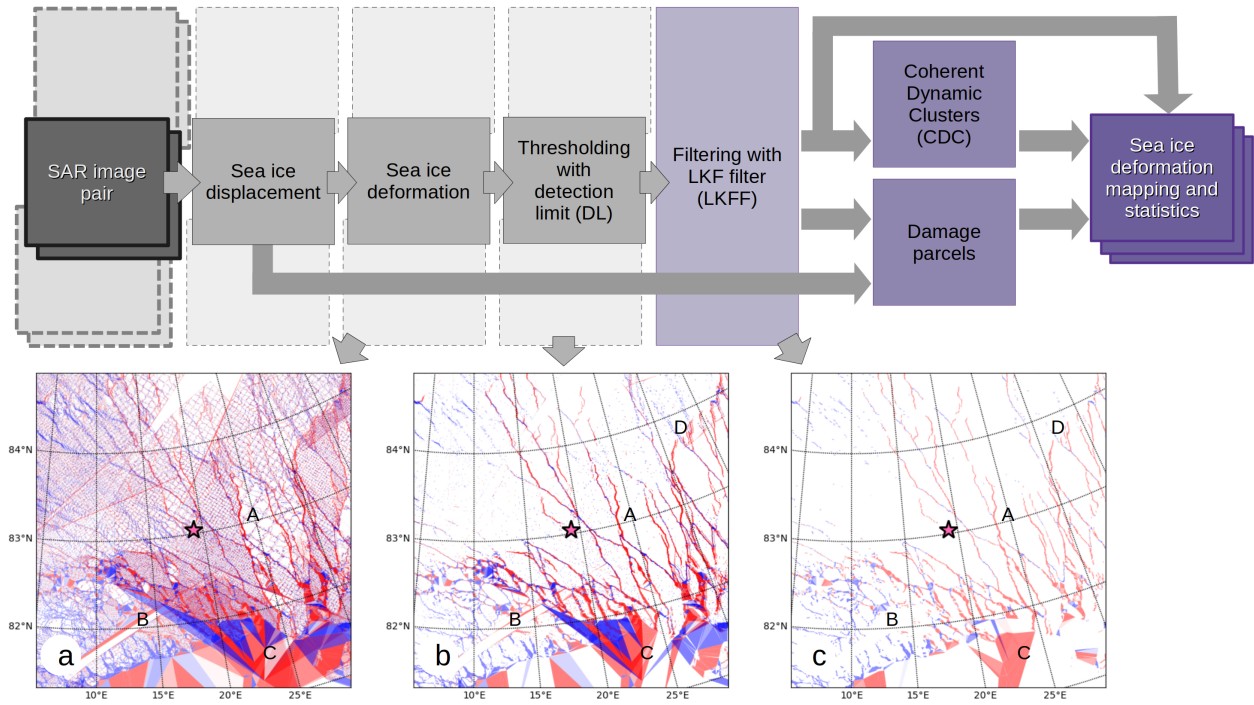

**Figure 3.** Flow chart of the SAR processing methods for sea ice deformation applied in this paper. 'SAR image pair' in the dark grey box is the input information. Light grey boxes show the intermediate steps with calculations of the sea ice displacements, deformation, DL thresholding and LKF filtering. Purple boxes show the outputs, each next level with an increase in color shading density. The secondary rows of the first four squares depict tiling. The LKFF products merges the tiles and spans the entire region. The bottom of the figure shows the examples of the SAR SAR deformation processing: a) original, b) DL filtered, and c) LKF filtered (LKFF) for a mosaic of SAR image pairs for 25 January 2015. The sizes of the frames are 400km by 400 km. The location of RV Lance is represented by a pink star in the centers of the frames. The step-wise evolution of specific cases is marked by capital letters: A – removal of rhomboids, B – removal of image edges, C – persisting problems in the MIZ , and D – merging of the tiled information ('disappearing shadow effect').

## 3.2 Comparison of the N-ICE2015 strain rates

To asses if the sea ice deformation derived from SAR by the methods described in the previous section was realistic, SAR-derived data was compared to the ship radar and buoy data. Unfortunately, were the ship radar and buoy deformation data used in this study just collections of values with no information about their spatial distribution. The ship radar deformation maps calculated by Oikkonen et al. (2017) were not geolocated nor stored and the spatial scales $\lambda$ of the buoy array are too coarse to compare directly to the ship radar values. To enable the comparison between all three datasets, the power law across all available $\lambda$ for of daily total deformation $\epsilon_{TOT}$ were estimated. Sea ice deformation is known to be a highly localized

process, where mean $\epsilon_{TOT}$ ($\bar{\epsilon}_{TOT}$) follows the power law with respect to $\lambda$: $\bar{\epsilon}_{TOT} = \alpha \lambda^{\beta}$ (Marsan et al., 2004), where $\alpha$ is the interception at $\lambda = 1$ and $\beta$ is the slope of the power law. The $\epsilon_{TOT}$ depends also on the temporal scale at which they were

measured - there is a power law also at the increasing temporal scale (Oikkonen et al., 2017). To account for this, the daily resolution of the buoy and ship radar data was strictly observed, while for the SAR image pairs a softer criteria was necessary and only temporal difference between 22 and 26 hours was used for this part of the analysis. The ship radar and buoy data was also filtered by $DL$ values determined by the SAR data. Because the data was collected over the same region and over the same time window, same magnitude of sea ice deformation was expected. This allowed a comparison of $\alpha$ in addition to typically used $\beta$ (Marsan et al., 2004; Bouillon and Rampal, 2015; Itkin et al., 2017; Oikkonen et al., 2017). The confidence envelope of the power law was estimated as a two-tailed T-test as in Itkin et al. (2017).

## 3.3 Damage Parcels

The spatially distributed sea ice deformation values obtained in the previous section were used to classify the sea ice cover into damaged (deformed) and undamaged (undeformed) ice. For such classification a zero threshold of total deformation was used. Afterwards, divergence values were used to further classify damaged ice into predominantly ridged (convergence) and predominant lead ice formation (divergence). The advantage of this two-stage classification was that first the total deformation was used to give a relatively reliable division between undamaged and damaged ice. Although the LKFF removed some of the spurious divergence, the second-stage classification was less certain.

Such classification was tracked for a sequence of time steps (SAR image pairs). The second SAR scene in each pair was always first scene in the following pair. The strict daily time difference between the image pairs was not observed. The tracking was not done for individual triangles, because of frequent distortion of the triangles beyond the 15 degrees minimal angle criteria. To avoid the numerically costly re-meshing, a simple approach with 'ice parcels' was used instead. Ice parcels can be used in applications where the mass preservation is not critical (e.g. Liston et al., 2020; Horvath et al., 2023). Displacements calculated as input to the sea ice deformation calculations were used to update the location of damaged ice parcels between the time steps.

At the beginning of the damage-tracking procedure the entire area is seeded by equally-spaced undamaged parcels. In subsequent steps each parcel is tracked and its value of damage, convergence or divergence accumulates by the deformation value found within the search radius. At the same time its location is updated. This procedure is repeated for all time steps. After the last time step the accumulated values of damage are used to classify each parcel into predominately leads, ridged ice, mixed class and undamaged ice. Figure 4 give an overview of the damage parcel tracking method. The spatial resolution of the damage parcel product in this study was 800 m, limited by the resolution of the input data (800 m spacing of the sea ice deformation and sea ice displacements), and by the search radius within which the input data is attributed to each parcel (also 800 m). If multiple triangle centroids fell within the search radius, the mean value was attributed to the parcel. If no value was found, the parcel disappeared. New parcels were added in the empty areas between the parcels i.e. wide leads. An early version of this method was used by Guo et al. (2022) for the same study area as in this paper.

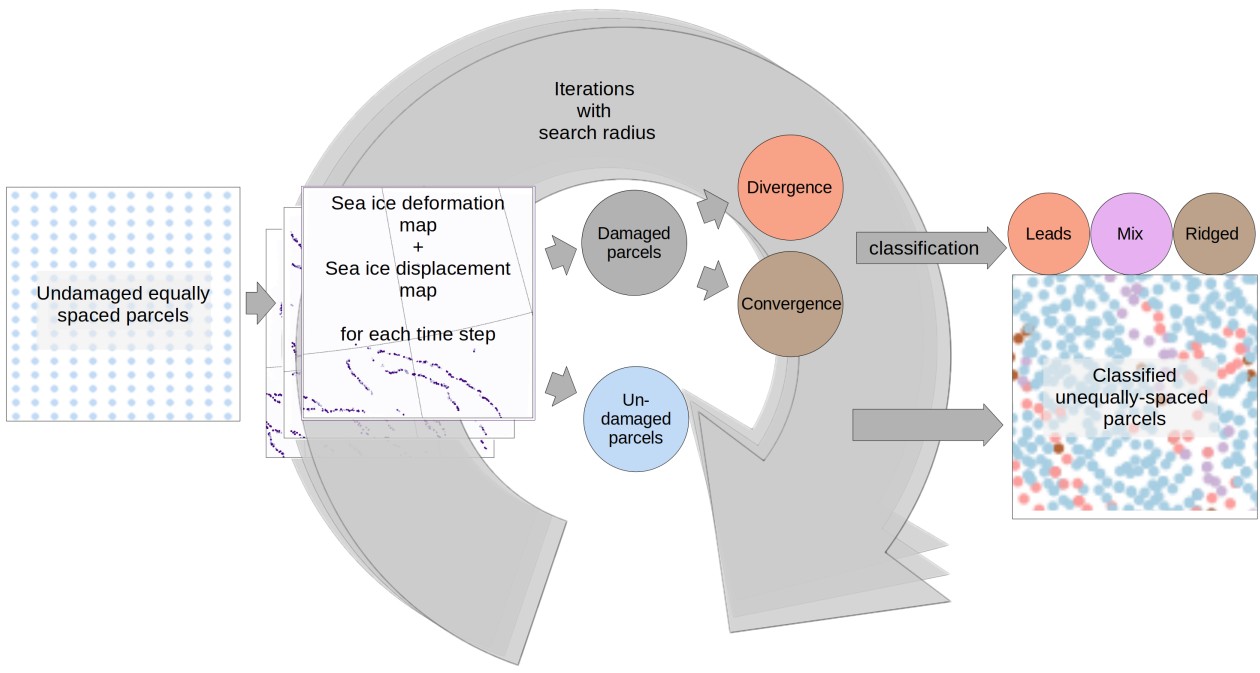

**Figure 4.** Flow chart of the damage parcel mapping applied in this paper.

## 3.4 CDCs and LKFs

As defined in the introduction CDCs are clusters of ice plates, containing damaged or undamaged ice, that at a given moment of time move coherently within each other and differentially in respect to the other CDC that are disconnected from them with fractures like LKFs. CDCs are transient features that only describe the state of ice pack in a certain moment of time. A statistical description of geometrical characteristics of CDCs and LKFs can therefore be used to evaluate the state of ice over time. Such statistical description suggested here was determined by using the location information of the deformation calculations. The triangle nodes filtered by LKFF were used to calculate polygons in computational geometry operations such as polygon union, distance buffering, difference (removed intersection) and removal of polygon holes as shown on Figure 5. The resulting polygons were classified into CDCs and LKFs and the following characteristics were determined at each time step:

– *CDC size* and *CDC density* are measured as area of the CDC ($km^2$) and number of CDC per total area $A$ ($km^{-2}$). At time steps with very low sea ice deformation the entire region may be divide into just one or a few large CDC, while at time steps with strong deformation it may divide into many small CDCs. CDC size is a variable more appropriate for mapping, while time series regional comparisons may be better achieved by comparing CDC density.

– *CDC circularity* is a measure of roundness and compactness of CDCs. It is determined by isoperimetric quotient - the ratio of CDC area $A$ and that of the circle having the same perimeter: $\frac{4\pi A}{p^2}$, where $p$ is the convex hull perimeter of CDC polygon. Completely circular CDC scores value 1, while smaller values are typical for less compact shapes (e.g. square score corresponds to 0.75). Convex hull perimeter is used instead of the CDE perimeter due to its sensitivity to overall elongation and angularity of the shape while being insensitive to the properties that are indicative of fragmentation. To compare the CDC shapes to the summer floe shapes values (e.g. Hwang and Wang, 2022), more frequently used *CDC roundness* was calculated. Roundness is less sensitive to 'squareness' as it is a simple ratio between max and min diameter of CDC (see next point), with values close to 1 for circular CDE.

– *CDC complexity* is a measure of fragmentation and it is determined by a ratio between CDC perimeter and its mean diameter. The mean diameter is the average of the min and max diameter estimated from min and max radius. First the centroid of the CDC is calculated. The min and max radii are then the shortest and the longest distance to the CDC boundary. CDC complexity has a theoretical value close to 2 for simple elongated polygons (practically a very slim rectangle). Typical values span from 4 for parallelograms expected for sea ice, and 7 for complex polygons with meandering boundaries. The latter is typical for CDCs where the LKFF likely fails to detect small deformation and correctly divide such into several smaller CDCs. Large CDC complexity is therefore a measure of error of the method.

– *LKF fraction* is LKF area per total area. It is determined in two separate methods: 1) by summing the area of of LKFF filtered triangles (min LKF fraction), and 2) by summing the area not covered by CDCs (max LKF fraction). The first method is more conservative than the latter, while the latter gives another measure of undetected deforming ice parcels.

– *Distance between LKFs* is estimated from the min and max CDC diameter used to estimate CDC complexity (see above). Based on the diameters min and max distance are estimated.

The vector operations required empirical tuning of parameters such as buffer distances and minimal sizes (Table 1). The buffer distance was used to merge disconnected polygons of the same LKFs in cases where parts of it were not detected. Later the same buffer was added to expand CDC polygons back towards the LKFs. The maximum LKF triangle size removed triangles in the MIZ, where the CDC and LKF detection is not possible. The minimal size of the CDC limited unrealistic fragmentation of ice cover into small CDCs.

## 4 Results and Discussion

### 4.1 Comparison of the N-ICE2015 strain rates

The spatial power law in $\epsilon_{TOT}$ was initially compared for the data obtained from the N-ICE2015 ship radar and buoys (Figure 6 a). For the complete dataset, the power laws exhibited similar intercepts, $\alpha$ (20.15 and 15.91), and slopes, $\beta$ (-0.6 and -0.69). Both power laws cover different spatial scales ($\lambda$): the ship radar data spans from 200 m to 3 km, while the buoy data spans from 3 km to 8 km. Before comparing them to the filtered SAR-derived data, the ship radar and buoy data needed to be filtered

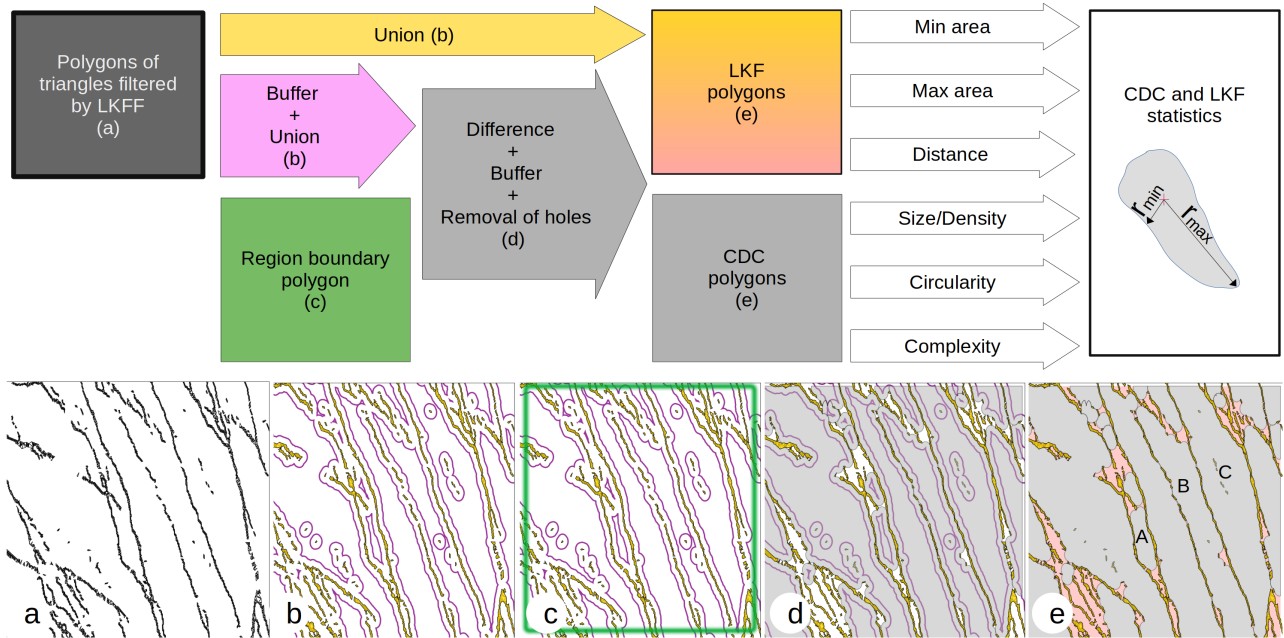

**Figure 5.** Flow chart of the CDC and LKF polygon operations and sub-figures showing examples of the sequential steps: a) union of LKFF triangles, b) 4-km buffers around the triangles and their union, c) regional frame that supplies part of the outer boundary to some of the polygons, d) difference between the region and the LKF polygons with removal of internal polygons (holes) and reclaim of the buffered area, and e) final CDC and LKF polygons. The step-wise evolution of specific cases is marked by capital letters: A – small initially not-separated CDC, B – initially partially disconnected boundary of a large CDC, and C – removed holes caused by under-detected LKF. The cartoon of a CDC in the box on the top right shows centroid (+), and min and max radii of polygon.

in a similar manner (Figure 6 center). Values below $DL$, estimated from mean $DL$ values in SAR data, were removed from the ship radar and buoy data. This filtering increased $\alpha$ for ship radar to 24.61 and deceased and $\beta$ to -0.7. For the buoys the removal of values below $DL$ led to a counter-intuitive slight decrease in $\alpha$ to 15.24. Note that the the power law for buoy data is based on mean values for only three $\lambda$ values, resulting in a wider confidence envelope. LKFF filtering of the ship radar and buoy data was not feasible, as described in the Method section under 'Comparison of the N-ICE2015 strain rates'.

Figure 6 (b) also shows the power laws for the SAR-derived data. The unfiltered data had a large number of low values that decreased the $\beta$ of the power law. DL filtering successfully removed these values and increased $\alpha$ from 4.79 to 14.63 and decreased $\beta$ from -0.14 to -0.59. To construct the power law, only means for $\lambda$ shorter than 10 km were used. This will be revisited later in this section. The LKFF filtering further increased $\alpha$ to 18.73 and decreased $\beta$ to -0.73. The power law from LKFF-filtered SAR data corresponded well to the ship radar and buoy data power laws within their confidence envelopes. The temporal resolution is not strictly daily for all datasets, and the area is not strictly the same, so some differences in power law

**Table 1.** Empirical parameters used in geometry operations to derive LKF and CDC polygons.

| name | description | value in this paper |
|---|---|---|
| buffer | Max distance (in m) between two nods of different triangles to get connected into same LKF. This is a circular buffer and there is a trade-off between desirable connecting over parts of a same partly undetected LKF and undesirable connecting across a breath of two separate, but close-by LKFs. | 4000 |
| LKF max triangle | Maximum area of LKF triangle to be taken into account. Here area of 12 right-angled triangles. This will exclude large triangles in the MIZ. | 3.84e6 |
| min area | Min area of the CDC and small CDC (in $m^2$). | 1e7 |

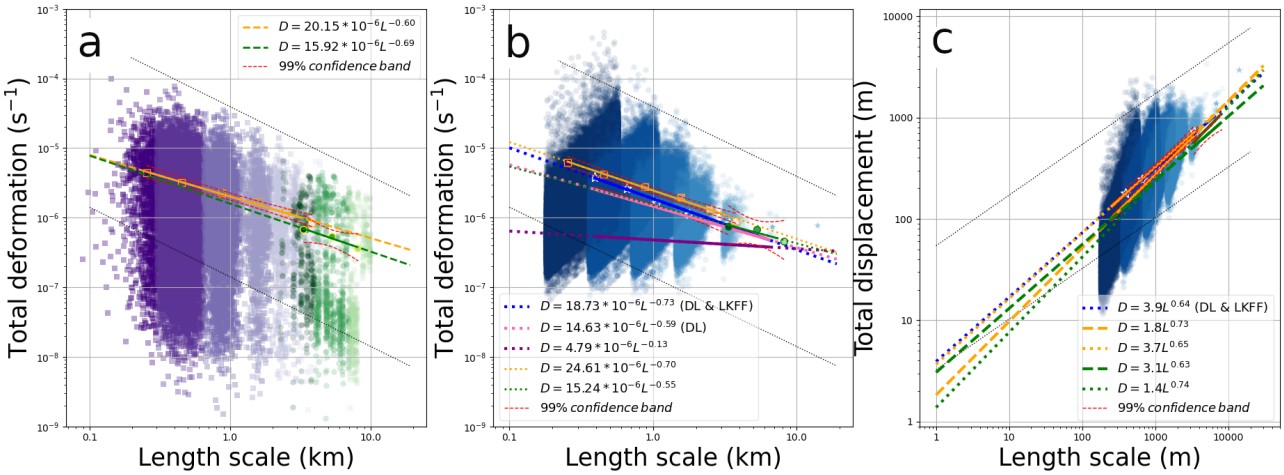

**Figure 6.** Power law for the spatial scales of ship radar-, buoy- and SAR-derived sea ice deformation calculations: a) original values from ship radar (purple shades with means for logarithmically distributed length classes as orange rectangles) and buoys (green shades with means as circles), SAR-derived deformation (blue shades with stars as means) and DL filtered ship radar and buoy values, and c) total displacements. Total displacements in panel c have a matching y-axis, but the x-axis to stretched to the left to show extrapolation of the power law to 1-m scale displacements. The power laws are fitted to the length-scale classes mean. Power laws with long dash lines include all data and the short dash lines power laws only include the DL filtered data. DL and max values are thin black dash lines on all plots.

are expected. Thus, the high deformation rates detected by SAR were reliable, and LKFF SAR-derived deformation was useful at short $\lambda$.

A further confirmation that the LKFF SAR estimates were reliable was the conversion of the $\epsilon_{TOT}$ values in Figures 6 (a and b) to mean total displacements ($\overline{\sigma_x}$). This was done by a simple rearrangement of Equation 5, used to estimate DL, such that $\overline{\sigma_x} = \sqrt{\frac{\epsilon_{TOT}(2Adt)}{3}}$. The displacements are more intuitive than strain rates and are comparable to any other distances over

space, such as ridge and lead widths or even sea ice thickness. A total displacement of 100 m can, for example, mean 70 m widening of a lead with 30m of shearing. Figure 6 (c) shows that $\overline{\sigma_x}$ at 1 km were about 200 m for all three datasets. This corresponded to a triangle with sides of about 1 km experiencing a sum of divergence and shear of approximately 200 m. If extrapolated to 100 m, $\overline{\sigma_x}$ was approximately 50 m. Both 1 km and 100 m values were within the expected ranges for medium and small size leads. At 1 m, $\overline{\sigma_x}$ for DL-filtered data were in the range of 3-4 m, and for the non-filtered data for ship radar and buoy data, it was in the range of 1-2 m. The latter values are comparable to the mean sea ice thickness measured in the pack ice at N-ICE2015 (Rösel et al., 2018). These values correspond to the expected theoretical values of $\overline{\sigma_x}$ (Weiss, 2017). A practical use of recalculating the strain rates to total displacement is also in observation (e.g., Nicolaus et al. (2022); Parno et al. (2022)) or simulation design. To resolve, for example, 100 m displacements (small leads), 10 m displacements (large ridges), or 1 m displacements (cracks), accurate measurements at 200 m, 30 m, or 1 m resolution at daily scale are required.

In addition to the power law of $\overline{\epsilon_{TOT}}$, other values scaled with $\lambda$ (Figure 6). For example, the DL values followed a power law which suggested that only at $\lambda \gg 10$ km, no $\epsilon_{TOT}$ values fell below DL anymore. This matched the noise-free estimates following Hutchings et al. (2012) (see the Method section). Finally, another sharp limit stood out in the data: the maximum values of $\epsilon_{TOT}$ and $\sigma_x$ follow a power law with a matching slope to the DL power law, but with an interception higher by approximately 2 orders of magnitude. At 1 km, the maximum $\epsilon_{TOT}$ and $\sigma_x$ values were approximately 40 1e-6 $s^{-1}$ and 2 km, respectively, for all three datasets. There is no natural explanation why $\sigma_x$ should be limited to values below 2 km, as Arctic leads and polynyas can be much wider than that (Wernecke and Kaleschke, 2015). This hard limit was a consequence of the 15° angle limit in the triangles used for strain rate calculations and another manifestation of the deficiency of estimates by line integrals (Green's theorem) (Marsan et al., 2004; Hutchings and Hibler III, 2008) in simple polygons like triangles. The 15° rule is often violated in fast-deforming features that have relative displacements beyond 2 km over a given time step.

The filtering of SAR-derived deformation resulted in widespread missing values (Figure 7). In the coarse-graining of $\lambda$, at least 10% of the area was required to have valid data. This means that as $\lambda$ became coarser, data had an increased fraction of missing values. The problem was demonstrated on maps in Figure 7 with an example from 25 January. The individual LKFs remained visible up to $\lambda$ 4.5 km. With $\lambda$ 9.6 km, the lines were discontinued, and only triangles containing either very wide or multiple LKFs at shorter $\lambda$ remained. The few remaining values at $\lambda$ larger than 10 km were relatively high, and their means did not follow the power law (Figure 6 b). Such a 'breaking point' between 5 and 10 km was previously detected in the comparison of the hourly buoy and ship radar data from N-ICE2015 (Oikkonen et al., 2017). The reason for it may reside in the under-detection of the self-similarity of sea ice deformation by the N-ICE2015 data. None of the datasets had a spatial resolution high enough to resolve the cracks. Likewise, all have a spatial extent limited to the local ice cover with a radius of 10 to 200 km, insufficient to resolve the pan-Arctic systems of LKFs. LKFs remain the only spatial feature which all three datasets could resolve. Figure 6 showed how removing low values increased the power law slope. Adding values from unresolved cracks would, therefore, decrease it. Some datasets that could be used to measure deformation cracks (resolution 1-m) already exist (Clemens-Sewall et al., 2022), but have very poor temporal resolution (weekly). On the contrary, adding the spatial extent should decrease the values at large $\lambda$ to prevent 'the power law breaking'. SAR data are already available at

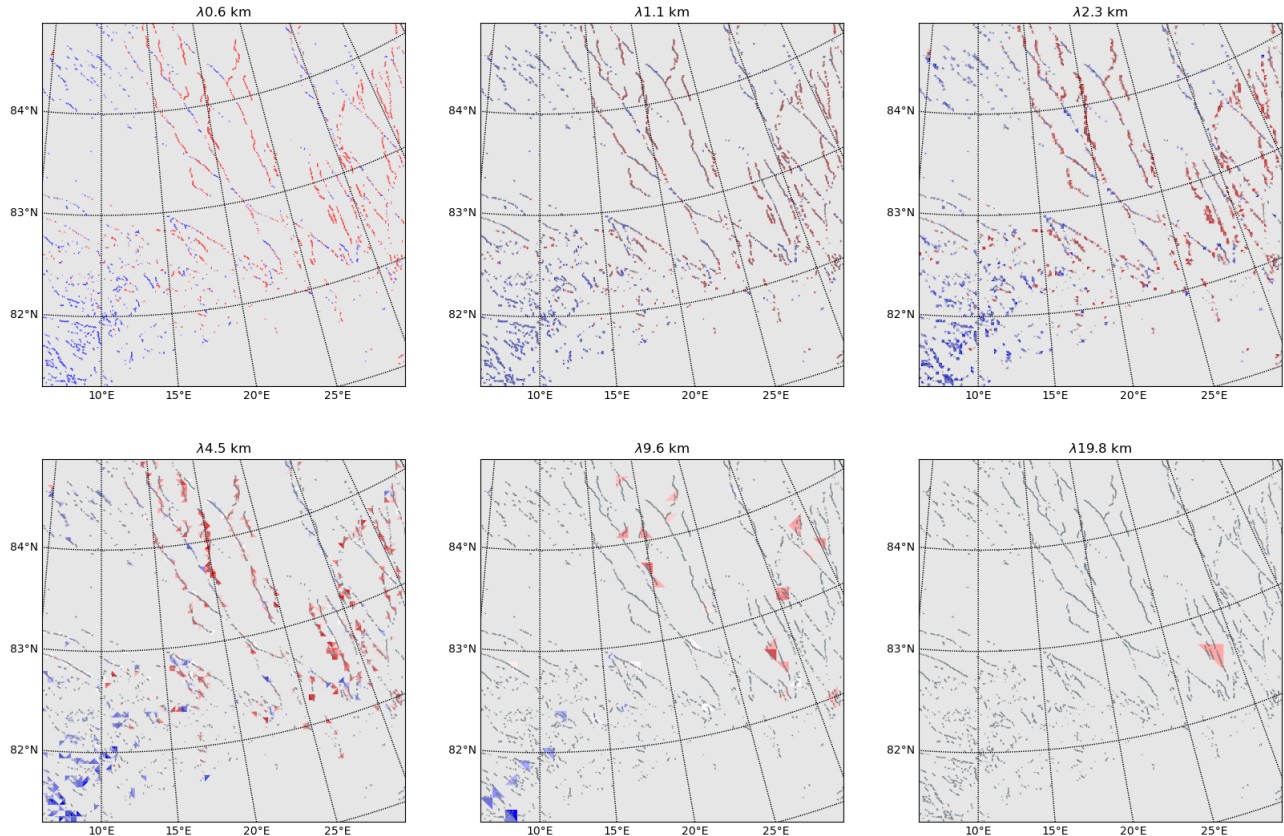

**Figure 7.** Example of coarse-graining of LKFF sea ice deformation for SAR image pairs for 25 January 2015. The sizes of the frames are 400km by 400 km. The area with values bellow DL is shaded in grey. The coverage of smallest triangles at $\lambda$ 0.6 km is shown on all other panels with dark grey triangle outlines.

the pan-Arctic scale (Kwok, 1998; Howell et al., 2022), but the resolution used in this study produces data volumes that are challenging and will be subject to future studies.

## 4.2 Damage Parcels

Sea ice damage parcels were tracked for 19 days, starting from 15 January to 3 February (see Figure 8). Daily image pairs were available from 15 January to 27 January. Subsequently, deformation was successfully calculated for an image pair from 27 January to 3 February, allowing the continuation of parcel tracking. During the 8-day period of missing data, from 27 January to 3 February, the conditions were relatively quiescent in terms of weather and sea ice motion (Itkin et al., 2017; Graham et al., 2019). Afterward, between 3 February and 5 February, SAR data was available, but the ice cover experienced a major storm (Graham et al., 2019), leading to unusually strong deformation (Itkin et al., 2017; Graham et al., 2019), which prevented further parcel tracking.

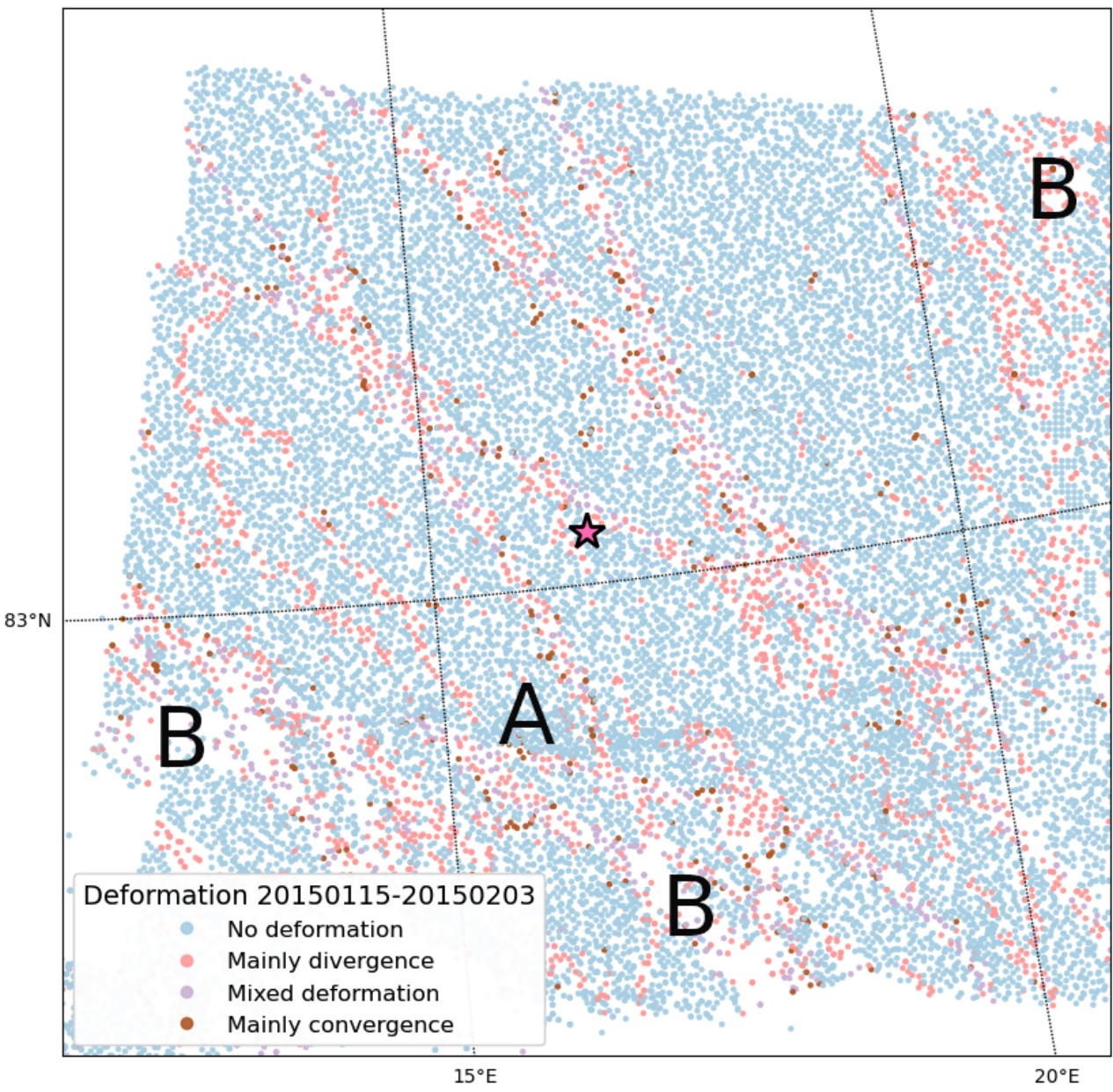

**Figure 8.** Damage parcel map after 19 days of sea ice drift between 15 January and 3 February 2015. The special regions of interest are marked by capital letters: A – locations with dense parcels indicating undetected convergence, and B - locations with sparser parcels indicating undetected divergence. The size of the map is 120 km by 120 km. The location of RV Lance is represented by a pink star in the center of the map.

The parcel map (Figure 8) illustrates a network of ice parcels primarily experiencing divergence within LKFs. This pattern persisted throughout the entire period, with the same parcels undergoing repeated deformation in the form of both divergence and convergence. Convergence and mixed deformation are localized and confined to narrow belts on the edges of predominantly diverging ice, a phenomenon observed despite changes in weather conditions and frequent alterations in sea ice drift directions (Graham et al., 2019). This finding aligns with previous studies and indicates that large fractures, where shear and shape mismatch occurred, healed slowly (Coon et al., 2007; Oikkonen et al., 2017). Recent observational studies on pressure ridges have confirmed that their keels consolidate over a period of weeks and months (Salganik et al., 2023a). The duration of sequential deformation observed in the damage parcel data can be valuable for constraining the damage-healing behavior of numerical models (Rampal et al., 2016; Damsgaard et al., 2021; Ólason et al., 2022).

Undamaged ice that did not experience any detected deformation did not move uniformly, resulting in variable distances between individual parcels (Figure 8). Although sea ice parcel classification was based on the filtered sea ice deformation retrieval, their motion was determined using the unfiltered sea ice displacement. The density of parcels in Figure 8 indicated undetected deformation with values below the $DL$. The noise originating from artifacts in the sea ice displacement algorithm was inherently random, and any regular spatial features visible in the ice parcel density were likely a consequence of deformation. For example, in the southern part of the map, there was a continuous curve with increased parcel density (marked by 'A' on Figure 8), which pointed to real deformation. Under convergence, the relative motion is typically shorter than under divergence and more challenging to detect. As another example, the areas in the southwestern and northeastern corners of the map had sparser parcels (marked by 'B' on Figure 8), indicating divergence.

## 4.3 CDCs and LKFs

In contrast to the damage parcels, CDC analysis was successfully applied to the sea ice deformation calculations over the entire period from January 15 to February 18. CDC maps (Figure 10 top) showed evolving shapes of the CDCs resembling ice plates seen in optical images (Erlingsson, 1988; Schulson, 2004). These shapes were obtained from daily SAR image pairs as well as those with shorter or longer time differences. The method was also successfully applied in the marginal ice zone (MIZ), albeit the method fails in the areas directly at the ice edge and open water areas. The maps displayed a gradient in CDC size, fragmentation, and roundness in the north-south direction from the pack ice towards the MIZ.

Based on the CDC maps (Figure 9), time series of CDC statistics (Figure 10) were calculated. The time series follow the drift of RV Lance to the ice edge until 22 February, after which they were extended by 10 days to account for the relocation of RV Lance northwards away from the MIZ (Granskog et al., 2018). In the first and the last week of the time series, statistics fluctuated around values typical for the quiescent state of the pack ice. The values changed abruptly with each of the three major storms (marked by A, B and C and Figure 10a) that passed the area (Graham et al., 2019):

– *CDC Density*, Figure 10a: The CDC density increased from low values below 5 CDCs per 100 by 100 km during quiescent periods to as high as 20 during the storms.

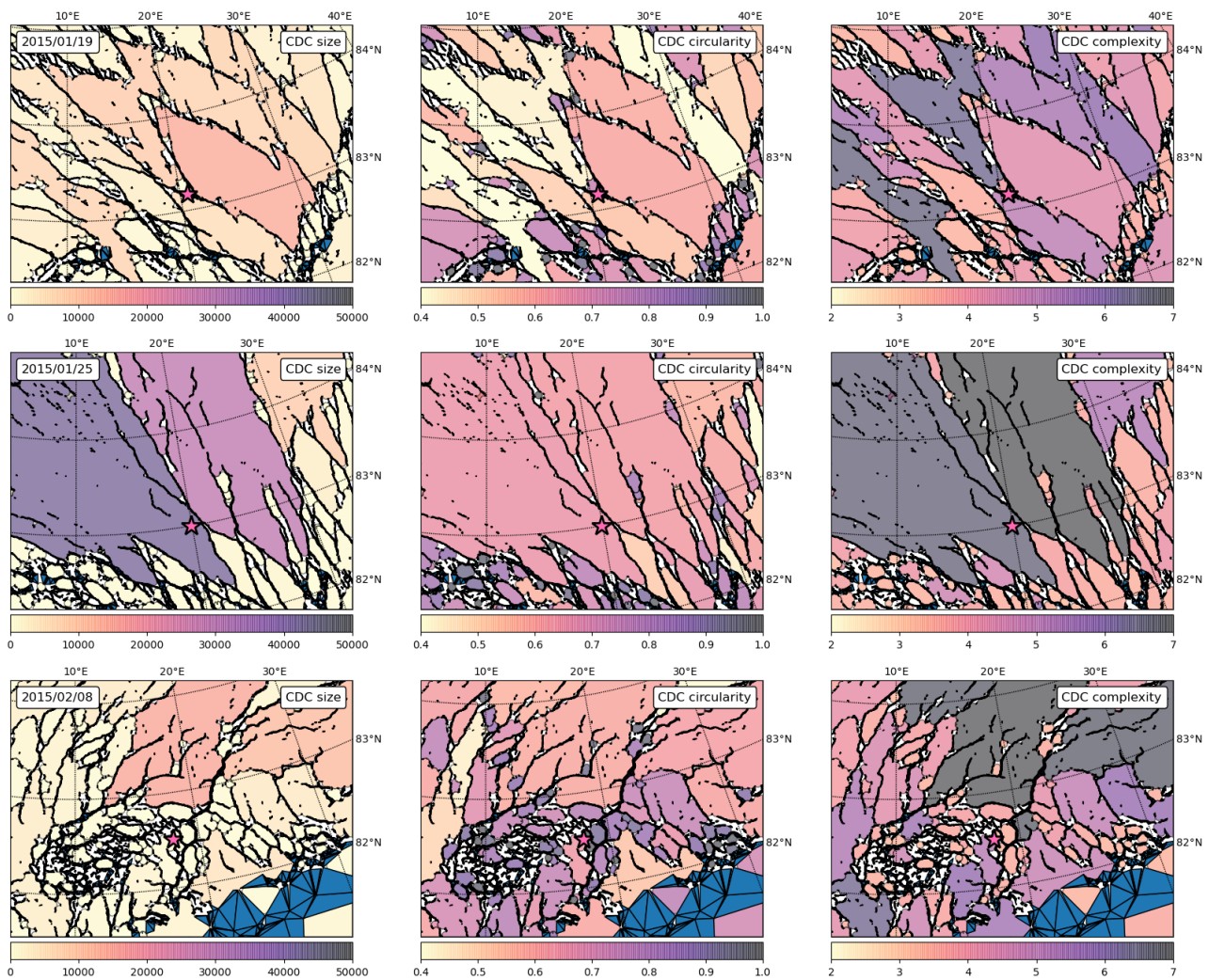

**Figure 9.** Examples of CDC maps with shape statistics (size, circularity and complexity) for SAR image pairs for 19 January, 25 January and 8 February 2015. Time stamps of the second image in the par is attributed to the result. The sizes of the frames are 400km by 300 km. The location of RV Lance is represented by a pink star. The distance from ship to MIZ is decreasing with time and on 8 February MIZ is already taking whole south-east corner of the map.

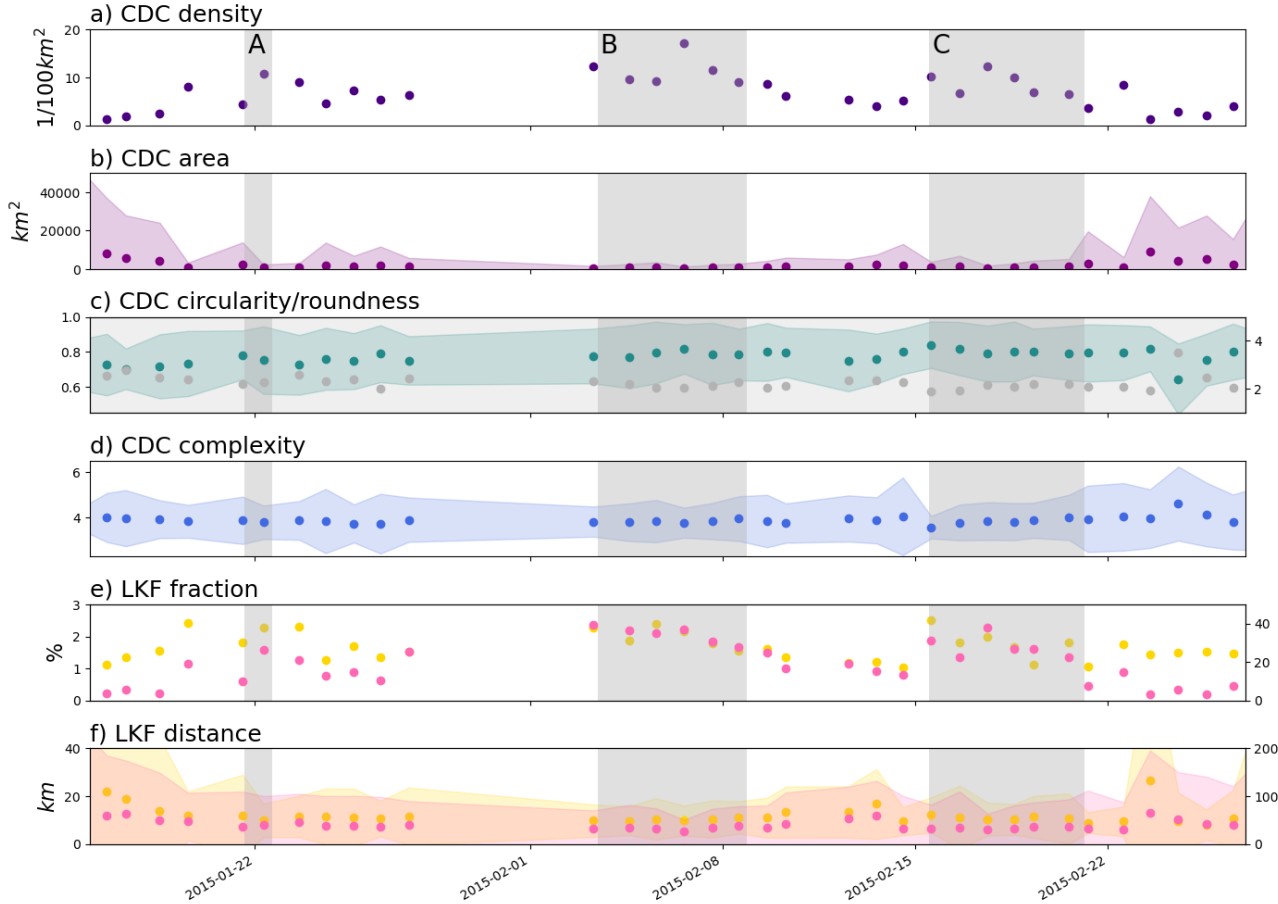

**Figure 10.** Time series of CDC statistics for N-ICE2015 leg 1 extended to 27 February 2015: a) density, b) area, c) circularity in green and roundness in grey, d) complexity, e) min LKF fraction in yellow and max LKF fraction in pink, and f) min distance between LKFs in yellow and max distance between LKFs in pink. The shading always implies one standard deviation from the mean. Storm periods are marked by grey shading and annotated by capital letters A, B and C. The ship relocated away from the MIZ on 22 February.

– *CDC Area*, Figure 10b: During the initial quiescent period, the number of CDCs was low, coinciding with a small variability in CDC area, with means just below 10000 km$^2$. After the first storm in mid-January (A), the values decreased by an order of magnitude to 1000km$^2$. The initial high values were never restored but increased significantly again after the relocation of RV Lance away from the MIZ.

– *CDC Circularity*, Figure 10c: The CDCs were generally not round, as their values lower than the theoretical value of 1 for circular forms. Typical values were closer to 0.75, indicating more rectangular forms. There was a continuous weak increase in CDC roundness during the entire duration of the time series. The *Roundness* of CDCs was typically about 2. In comparison, typical roundness values of summer sea ice floes are simlar - between 1 and 4, but with the most frequent value reported as 1.4 (Hwang and Wang, 2022).

– *CDC Complexity*, Figure 10d: This measure of under-detected fractures exhibited the largest variability during quiescent periods, reflecting the smallest and most challenging relative motion to detect by SAR. The maps showed that fragmentation was lowest in the MIZ, where distances between the CDCs were largest, and relative motion was high (Figure 9).

– *LKF Fraction*, Figure 10e: The lowest LKF fractions, close to 1%, were recorded in mid-January and occasionally increased towards 3% during storms. They remained relatively high (close to 2%) in quiescent periods and after the relocation away from the MIZ. The max LKF fractions were typically above 5% during quiescent periods, but as high as 30% during the storms. This indicates that the clear definition of CDC boundaries during the storms is difficult.

– *Distance between LKFs*, Figure 10f: The maximal and minimal distances between LKFs estimated from CDC diameters were typically at least 5 to 10 km and at most 50 to 100 km. The highest values were found during quiescent periods and were lowest during the storms. The distance between the LKFs derived from the CDC perimeters confirmed the findings about the power law 'breaking,' discussed in the Results and Discussion section under 'Comparison of the N-ICE2015 strain rates.' The lowest distances were between 5 and 10 km, corresponding to the $\lambda$ breaking point.

Despite the fierceness of the storm (B) in mid-February (Itkin et al., 2017; Graham et al., 2019), the CDC properties at the end of the time series resembled the ones at the beginning. This pointed to the resilience of the CDCs and their geometrical properties in the winter pack ice and the potential for healing. However, some properties such as CDC area, circularity, and LKF fraction never fully recovered to the values recorded at the beginningof the time series. Itkin et al. (2017) previously demonstrated an elevated slope of the power law of the N-ICE2015 spring buoy array (April-May 2015) and hypothesized a long-lasting damage to the sea ice cover caused by this extreme storm.

## 5   Conclusions

This study has successfully demonstrated that Sentinel-1 SAR-derived sea ice deformation data can be effectively utilized at a spatial resolution of 800 m after mitigating noise generated by sea ice displacement algorithm. This was accomplished through

the application of threshold and shape filtering techniques. The accuracy of the SAR-based results was validated against data from the N-ICE2015 ship radar and buoy observations, utilizing power law properties.

Across spatial scales ranging from 600 m to 5 km, the power law slope and intercept for all three datasets (processed in a similar manner) were consistently close to 20 and -0.7, respectively. Subsequently, the strain rates were recalculated to obtain total displacements, revealing a mean total displacement of approximately 200 m at the spatial scale of 1 km.

While the power laws of all three datasets exhibited a strong correspondence below 5 km, the SAR-derived power law 'breaks' for the larger spatial scales. This discrepancy was likely attributed to issues related to spatial resolution, wherein fractures smaller than LKFs are inadequately resolved, coupled with limitations in spatial extent.

The detection of SAR deformation was constrained not only by the lower limit of spatial resolution but also by the minimum angle of triangles employed in Green's theorem (upper limit).

The damage parcels, derived from SAR-based displacements and strain rates, were effectively employed to monitor the evolution of the sea ice cover over a period of three weeks, encompassing the passage of a major storm and an 8-day data gap. The temporal stability of the ice pattern was disrupted by an extreme storm in mid-February, as reported by Graham et al. (2019), after which the tracking of damage parcels became unfeasible.

Up to 20 separate CDCs per area of 100 km by 100 km for each SAR image pair were identified based on the positions of LKFs detected through SAR-derived deformation. During quiescent periods, their typical sizes were 10,000 $km^2$, whereas they reduced to 1,000 $km^2$ during storms. These CDCs exhibited mainly rectangular shapes, more elongated than the sea ice floes after breakup, as reported byHwang and Wang (2022). The fraction of the surface area covered by LKFs remained typically below 3%, but the boundaries between CDCs and LKFs may become difficult to establish during the strongest deformation events. The distances between LKFs varied between 5-10 km and 50-100 km, reflecting minimal and maximal values, respectively.

All the methods employed in this study are applicable to both the winter sea ice pack and the MIZ areas. Furthermore, the spatial resolution can be enhanced for Sentinel-1 SAR, and these methods remain adaptable to dense sea ice deformation retrieval from radar or optical imagery, regardless of the data source. In the follow-up paper the method will be employed to the longer time series over the entire winter season (e.g. MOSAiC). This will allow for the analysis of the processes such as 'reactivation' of damage parcels after a long 'relaxation time' and seasonal development of CDCs.

*Data availability.* TEXT

Sentinel-1 SAR data are freely available and were downloaded from CREODIAS (https://creodias.eu/).

*Code and data availability.* TEXT

The code used in the manuscript is written in Python (with dependence on functions in libraries NumPy, SciPy, PyResample and Shapely) and it is available in the GitHub repository (https://github.com/loniitkina/sid/). The version used to obtain the results of this study will be available at Zenodo at the end of the review process.

*Author contributions.* TEXT

PI processed the data, carried out the analysis and wrote the manuscript.

*Competing interests.* TEXT

PI declares that she has no conflict of interest.

*Acknowledgements.* PI is especially grateful to Jari Haapala for fruitful discussions and encouragements over many years working on the data and developing the methods. Discussions with Gunnar Spreen and Anton Korosov and many conference and workshop attendants are

likewise acknowledged. PI is also grateful to Annu Oikonnen for providing the ship radar strain rates. Andy Mahoney and one anonymous reviewer were the first to read this work and provided excellent suggestions for improvements. The TC editor team with Vishnu Nandan was very supportive during the lengthy way towards the publication - the author takes all responsibility for the slow process. The work for this paper was funded by the Research Council of Norway project SIDRiFT (#287871). This work would have not been possible without watching sea ice deformation live during N-ICE2015 and MOSAiC expeditions.

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
