# Peer review of "Novel methods to study sea ice deformation, Linear Kinematic Features and Coherent Dynamic Clusters from imaging remote sensing data"

_EGUsphere, 2023_

## Referee Comment (RC2)

**Novel methods to study sea ice deformation, linear kinematic features and coherent dynamic elements from imaging remote sensing data**

by Polona Itkin

*Submitted to Cryosphere Discussions*

**Review**

*By Andy Mahoney*
*Dec 21, 2023*

**Summary**

This manuscript presents an analysis of sea ice deformation derived primarily from analysis of sequential synthetic aperture radar (SAR) images acquired north of Svalbard during the N-ICE2015 expedition. The novelness comes from the identification, naming, and quantitative characterization of coherent dynamic elements (CDEs). These features exist as regions of uniform ice motion between linear kinematic features (LKFs), which I believe were first described by Kwok in 2001. Ice motion is tracked through a combination of feature tracking and pattern matching to create a grid of regular spaced nodes, which are triangulated for the calculation of deformation. LKFs emerge as linear clusters of triangles that experience high degrees of shear, divergence, or both. CDEs occupy the regions between LKFs and the manuscript uses a number of different shape-related metrics to chart their evolution over the course of three separate storm events.

The manuscript also presents a scaling analysis of strain rate by combining the SAR-derived deformation measurements with those made using the ship radar and a network of GPS-tracked buoys. This allows power laws for strain rate and total deformation to be evaluated for length scales ranging from hundreds of meters to tens of kilometers.

Overall, I enjoyed reviewing this manuscript. I believe the CDEs that the author defines represent a useful concept for framing our thinking about the concept of ice floes and how stress and strain are organized within the ice pack. I have no significant concerns, but I feel the analysis of CDE shape statistics could be usefully improved and some more attention could be given to the structure and standardization of the text and figures to aid readability. I have tried to provide constructive suggestions with each of my comments below and I do not believe any of my concerns should be difficult to address.

**Major Comments**

**1       Methods section would benefit from restructuring**

I feel that readability of the methods sections could be improved with some restructuring of the text. Specifically, I would encourage the author to adopt a more chronological approach with description of the different steps involved in deriving the various results. For example, I find it highly non-intuitive that the text in section 3 goes into significant detail regarding the derivation of deformation components from triangulated ice motion data before the method for determining ice motion has been described in section 3.1.

On line 87, the text states "*information from individual images was processed into displacements, drift, and deformation*" and I think this would be a more appropriate order in which to describe the methodology to the reader (i.e., displacement and drift before deformation). However, at the same time, I also recommend reviewing the manuscript for consistent use of these terms. In particular, I cannot determine the difference between "drift" and "displacement" and the terms appear to be used

interchangeably later in the text, which makes it a little confusing to list them separately here. I recommend choosing one term and using it consistently.

**2     More detail needed regarding definition of damage parcels**

I think I understand how the damage parcels are defined, but I had to read section 3.3 multiple times to do so. For me, it would help to explicitly clarify the relationship between the location and distribution of damage parcels and the nodes of the triangulated grid. I'm a visual thinker and I believe many other readers are too, so I would encourage the use of an additional figure to help crystalize this important concept.

**3     Clearer definitions of spatial scales**

On line 144, the text equates a value of "$\lambda \gg 679\ m$" with "*scales approximately 10 km*" (sic). I feel more explanation is needed here to explain how "scale" is defined here and how it is quantitatively related to the value of $\lambda$.

**4     Definition and use of CDE shape parameters could be improved**

I have a number of comments about the definitions and explanations of the CDE shape parameters used in this manuscript. These are detailed in the sub-comments below, but in short, I recommend that the author considers alternative, potentially more suitable statistics and define them in more detail in the text. In particular, I encourage the use of a figure to help explain some of the details.

**4a     *Shape diameters and radii**

The bulleted text in section 3.4 makes a number of references to the diameter or radius of CDEs, but does not provide the reader with the specific details necessary to understand how these terms are applied to highly non-circular shapes like those of many CDEs. From context, I expect the "*max and min diameter*" used in the definition of roundness (line 236) is measured using either caliper distances or the dimensions of an enclosing rectangle, though in either case these are not strictly diameters since they do not necessarily pass through the center of the shape. Similarly, the "*mean diameter*" used in the definition of fragmentation (line 233) is not sufficiently defined. It could be the average of the minimum and maximum diameters, though a more common definition is the diameter of a circle with equal area to the shape in question. There are also other definitions of diameter and radius that would require explicitly defining the center of the CDE.

**4b     *Fragmentation parameter**

I had difficulty understanding how the definition of this parameter on lines 233-236 could be related to fragmentation of CDEs until I looked ahead to Figure 9 (hence why I recommend using an additional figure at this point in the manuscript). As defined, it is a measure of the complexity or non-circularity of the CDE perimeter, which does not necessarily have anything to do with fragmentation. An alternative statistic that might be more sensitive to how far the perimeter penetrates into the interior of the CDEs is the convexity, which can be defined as:

$$concavity = \frac{convex\ hull\ perimeter}{CDE\ perimeter}$$

This will take a value of 1 if the shape is convex or less than one if the perimeter has concavities. One advantage of this metric is that there is no underlying reliance on the properties of a circle. Also, there is a possible variant of this approach whereby the perimeters of all LKFs within convex hull (or just their skeletonized lengths) are also included with the CDE perimeter in the denominator. This would take account of all the unconnected LKFs that are currently encapsulated within CDEs without contributing to the fragmentation statistic.

*4c      Roundness parameter*

As defined, this parameter could just as well be called "squareness" (a square would give the same value as a circle), but the parameter is really a measure of aspect ratio or elongation. A more suitable measure of roundness or circularity could be the ratio:

$$circularity = \frac{4\pi \cdot CDE\ area}{(convex\ hull\ perimeter)^2}$$

By using the convex hull perimeter (instead of the CDE perimeter), this metric is sensitive to overall elongation and angularity of the shape while being insensitive to the properties that are indicative of fragmentation.

*4d      Distance between LKFs*

The distance between LKFs is defined as "*min and max CDE radius*" (line 239), but I suspect this should read "diameter" instead of radius. Please also see comment 4a above regarding detailed definitions of such dimensions.

5      Figure sub-panels

This is a minor comment, but applies it applies to every multi-panel figure. I encourage the author to label each sub-panel with a letter and use this an identifier in the text instead of relative descriptors like "left" and "right". I understand this is partly stylistic, but I feel it removes any ambiguity when cross-referencing sub-panels both in the caption and in the main text (e.g., lines 341 and 346). It may also be required by the journal.

**Minor comments**

Line 144: I think there's an "*of*" missing before "*approximately*".

Line 153: I recommend adding letters to identify each panel (see comment 5 above). Although the references to "left", "right", etc in the caption are technically accurate, I find it would be helpful to explicitly identify that each instance references more than one panel in the figure.

Line 162: Replace "*of*" with "*by*"

Lines 167 & 168: I assume the author intended to remove these question marks before submission. Do they signify missing citations?

Line 170: What is "*DLDL*"? Is this simply a typo for DL?

Lines 170-171: I feel a little more explanation of Bouillon and Rampal's smoothing technique is necessary here. I do not think it should be necessary to be familiar with a cited work to understand the text in which the citation is made. Also, in this case, the text refers to a "kernel size of 3 triangles", whereas Bouillon and Rampal define the size of their kernel in terms of the number of vertices.

Figure 4: In its current form I do not feel this figure gives the reader much more information than is already provided in the text. Elsewhere, I recommend the inclusion of additional figures (see comments 2 and 4 above), so if space is an issue this figure could possibly be omitted. However, if the authors choose to keep the figure, then I feel some explanation should be given for the different colors used for boxes and arrows.

Line 209: Insert "*each*" between "*in*" and "*pair*"

Line 213: There are some unnecessary parentheses in this citation

Lines 214-215: See comment 1. Are displacements and sea ice drift the same thing?

Line 223: Replace "*nods*" with "*nodes*"

Line 233: If the author chooses to keep their definition of fragmentation (see comment 4b above) then I would replace "circumference" on line 233 with "perimeter", since the term circumference is only defined for a circle. Also, according to the definition given, a perfect circle would have a fragmentation value of 2π, not 0.

Figure 5: Similar to my comment for Figure 4, I do not feel figure 5 adds much value to the text. Instead, I think I would find it more useful to see a graphical representation of how nodes associated with LKFs and CDEs are enclosed in convex hulls (see comment 4 above).

Figure 6: The axis labels are quite difficult to read and the neither the caption nor the legend explain the meaning of the different colors. Also, I recommend using the same units and limits for the x-axes of all three plots.

Line 248: I think "*similar*" may be a more suitable word than "*resembling*"

Line 249: Replace "ad" with "and"

Lines 252-253: The text "*This increased $\alpha$ and $\beta$ for ship radar to 24.61 and -0.7, respectively*" is confusing to me. The value of $\alpha$ is greater than that reported earlier in the paragraph for the full suite of data, but value of β represents a decrease. Some rewording may be necessary here.

Lines 256-265: This paragraph appears to be a duplicate of the preceding paragraph.

Line 266: Replace "amount" with "number", since this refers to a countable quantity.

Lines 267-268: This could be re-written as "*... increased $\alpha$ from 4.79 to 14.63 and $\beta$ from -0.14 to -0.59*" to both improve clarity of the text and remove the need to use "respectively". This practice could be adopted in other place in the text too.

Line 277: Specify units after "*100*" (presumably meters).

Line 278: I'm not sure what "*boarding*" means in this context. Is this a typo? I would recommend using either "opening" or "widening" instead.

Figure 8: I think I would understand and appreciate this figure more fully if I was confident I understood how the damage parcels were defined and located. Please refer to my comment 2 above.

Line 325: I assume this question mark was not intended to be included in the text, like those on lines 167 and 168.

Line 326: I do not think Murzda et al's paper supports the assertion that "*large fractures ... healed slowly*". First, Murzda et al report crack healing on timescales of "tens to hundreds of seconds", which I would characterize as quite rapid in this context. Second, I'm not convinced their lab-based observations can easily be scaled up to that of the "large fractures" described in this study. Murzda et al explicitly note this at the end of section 1 of their paper.

Line 335: The "*continuous curve with increased parcel density*" is a pretty subtle feature. I recommend labelling it on this figure with some form of annotation.

Figure 9: The significance of the gray regions in the time series plots is not explained in the caption. From indirect cues in the main text, I assume they indicate the occurrences of storms. In addition to explaining their meaning in the caption, I also encourage the author consider

naming and labelling each storm uniquely (e.g., "mid-January storm", etc, or more simply Storm 1, ...). This would allow easy and clear cross-referencing in the main text. I also encourage labeling each sub-panel with a letter, as per comment 5 above.

**References cited in this review that are not cited in the manuscript**

Kwok, R. (2001), Deformation of the Arctic Ocean Sea Ice Cover between November 1996 and April 1997: A Qualitative Survey, paper presented at IUTAM Symposium on Scaling Laws in Ice Mechanics and Ice Dynamics, Springer Netherlands, Dordrecht.

---

## Author Comment (AC1)

**Novel methods to study sea ice deformation, linear kinematic features and coherent dynamic elements from imaging remote sensing data**

by Polona Itkin

Submitted to Cryosphere Discussions

**Final response to reviewer Andy Mahoney**

I thank the review for his investment in the review, a very positive review and a large number of valuable suggestions.

Here are the major comments addresses one-by-one:

1. Methods section would benefit from restructuring

Here I understand that the structure is not ideal – since it confused the reviewer that is a sea ice deformation expert. However, instead of complete restructuring, I will introduce the noise limit calculation and the definition of displacement, drift and deformation in the beginning of the method section (3). This introductory part of the methods refers to all the data described in the Data section (2). The buoy data and ship radar require no specific methods novel to this paper. Section 3.1 is then listing the specifics of the novel SAR deformation calculations and is sequential. I will include statements to make this more clear in the manuscript.

2. More detail needed regarding definition of damage parcels

I will add a figure where I explain the sequence (in time) how the deformation is tracked on a map.

3. Clearer definitions of spatial scales

I agree. I will explain that 10 km is used as it is an order of magnitude larger than 679m. I review if the definition of lambda is sufficient.

4. Definition and use of CDE shape parameters could be improved

I really appreciate the input the reviewer made to this point! These are all excellent comments that will be taken into account and by doing this greatly improve the paper. I will also add a schematic (similar as under major comment 2) here with drawings of the diameters, distances etc.

5. Figure sub-panels

Yes, figure sub-panels will be marked with alphabetic indexes for the final version of this paper.

Minor comments:

All minor comments are relatively easy to adapt and will be taken into account.

---

## Author Comment (AC2)

**Novel methods to study sea ice deformation, linear kinematic features and coherent dynamic elements from imaging remote sensing data**

by Polona Itkin

Submitted to Cryosphere Discussions

**Final response to reviewer 1**

I thank the reviewer for his/her investment in the review, generally very positive review and many valuable suggestions.

1. CDE definition:
I will improve the definition of the CDEs and how they are introduced in the text. CDEs are the coherently moving ice plates. The characteristics of these CDEs can be described by their geometry, and this is where the architecture of the shape terms is introduced. The improvements of the definitions of these geometrical shape terms will be improved as suggested in the review by Andy Mahoney. Also the schematic figure explaining the geometrical properties and showing examples od CDEs will be helpful. Furthermore, I will summarize the winter pack ice characteristics based on the CDE geometries in the conclusions in a better way.

2. Power law:
I disagree with the reviewer on this point. Power law is 'the bread and butter' of the sea ice deformation comparisons. The scale differences (in space and time) between the datasets do not allow for any other comparison (and for ship radar they were not even stored). I believe it is good that this analysis is in a separate chapter that those who want can look at it in detail, while the rest of the readers can focus on the other parts of the paper. I will however, improve  also this text, so that it will be hopefully easier to read for everybody.

3. Lack of synoptic data:
The weather time series from Graham et al, 2018 (Scientific Reports) are available and could be used for interpretation of the N-ICE2015 data. They are the base of the shading on Figure 9. I will consider adding these synoptic data to Figure 9. If this will sidetrack the manuscript from the main purpose of the method description, all references to synoptic events will be removed from this paper.

4. Too many acronyms:
I will carefully review all the acronyms and see if any can be removed.

All minor comments are relatively easy to adapt and will be taken into account. Especially the link of the Abstract – Introduction – Conclusions will be reviewed to improve the readability of the paper and enhance its message.

---

## Author Response (AR1)

**Novel methods to study sea ice deformation, linear kinematic features and coherent dynamic elements from imaging remote sensing data**

by Polona Itkin

Submitted to Cryosphere Discussions

**Final response to Reviewer 1**

I thank the reviewer for his/her investment in the review, generally very positive review and many valuable suggestions.

1. CDE definition:

Reviewer writes:
I really like the idea of CDE's but their definition is a bit confusing.  It seems to me CDE's are an architecture or framework or terms (not term singular) that certain variables can be used to collectively describe winter pack ice. Am I right? However, you first define Coherent Dynamic Elements (CDE) as the boundary of rigid ice plates (Line 58 and 59). OK. In the Abstract you say CDE describes the behaviour of the winter pack but nothing in the paper including your Conclusion relates the winter pack behaviour during N-ICE2015 in that context. I thought I was missing something. Further, if a new term is introduced, then the definition must be consistent. Your definition and usage of CDE needs revision throughout the text otherwise readers will be scratching their heads as its meaning and usage. I suggest defining the CDE framework (with associated variables) earlier in the paper and explicitly describe how these terms can be used collectively to understand winter pack behaviour with evidence from N-ICE2015.

*I have improved the definition of the CDEs and how they are introduced in the text. The term has now been renamed to 'Coherent Dynamic Clusters' (CDC) and introduced in the introduction as:*

*'These fractures align along the Mohr-Coulomb failure lines, dividing the ice surface into distinct, parallelogram-shaped plates that move relative to one another, much like tectonic plates on a planet. \citep{erlingsson1988,shulson2004,dansereau2019}. In this paper the cohesive clusters of these plates are called 'Coherent Dynamic Clusters' (CDC) - a name that describes the transient nature of their motion along the fractures. The CDCs can be described by size and shape parameters, offering a novel option for sea ice deformation characterization.'*

*Then the CDC definition is briefly summed and method of their characterization further explained in the Methods section:*

*'As defined in the introduction  CDCs are clusters of ice plates, containing damaged or undamaged ice, that at a given moment of time move coherently within each other and differentially in respect to the other CDC that are disconnected from them with fractures like LKFs. CDCs are transient features that only describe the state of ice pack in a certain moment of time. A statistical description of geometrical characteristics of CDCs and LKFs can therefore be used to evaluate the state of ice over time. Such statistical description suggested here was determined by using the location information of  the deformation calculations. The triangle nodes filtered by LKFF were used to calculate polygons in computational geometry operations such as polygon union, distance buffering, difference (removed intersection) and removal of polygon holes as shown on Figure \ref{fig4}. The resulting polygons were classified into CDCs and LKFs and the following characteristics were determined at each time step:*
- *\textit{CDC size} and \textit{CDC density} are measured as area of the CDC ($km^{2}$) and number of CDC per total area $A$ ($km^{-2}$). At time steps with very low sea ice deformation the entire region may be divide into just one or a few large CDC, while at time steps with strong deformation it may divide into many small CDCs. CDC size is a*

- *variable more appropriate for mapping, while time series regional comparisons may be better achieved by comparing CDC density.*
- *\textit{CDC circularity} is a measure of roundness and compactness of CDCs. It is determined by isoperimetric quotient - the ratio of CDC area $A$ and that of the circle having the same perimeter: $\frac{4 \pi A}{p^2}$, where $p$ is the convex hull perimeter of CDC polygon. Completely circular CDC scores value 1, while smaller values are typical for less compact shapes (e.g. square score corresponds to 0.75). Convex hull perimeter is used instead of the CDE perimeter due to its sensitivity to overall elongation and angularity of the shape while being insensitive to the properties that are indicative of fragmentation. To compare the CDC shapes to the summer floe shapes values \citep[e.g.][]{hwang2022}, more frequently used \textit{CDC roundness} was calculated. Roundness is less sensitive to 'squareness' as it is a simple ratio between max and min diameter of CDC (see next point), with values close to 1 for circular CDE.*
- *\textit{CDC complexity} is a measure of fragmentation and it is determined by a ratio between CDC perimeter and its mean diameter. The mean diameter is the average of the min and max diameter estimated from min and max radius. First the centroid of the CDC is calculated. The min and max radii are then the shortest and the longest distance to the CDC boundary. CDC complexity has a theoretical value close to 2 for simple elongated polygons (practically a very slim rectangle). Typical values span from 4 for parallelograms expected for sea ice, and 7 for complex polygons with meandering boundaries. The latter is typical for CDCs where the LKFF likely fails to detect small deformation and correctly divide such into several smaller CDCs. Large CDC complexity is therefore a measure of error of the method.*
- *\textit{LKF fraction} is LKF area per total area. It is determined in two separate methods: 1) by summing the area of of LKFF filtered triangles (min LKF fraction), and 2) by summing the area not covered by CDCs (max LKF fraction). The first method is more conservative than the latter, while the latter gives another measure of undetected deforming ice parcels.*
- *\textit{Distance between LKFs} is estimated from the min and max CDC diameter used to estimate CDC complexity (see above). Based on the diameters min and max distance are estimated.*

*In addition, the flowchart has been greatly improved with some visualizations of intermediate steps.*

*Note that the descriptions of some CDC properties is now slightly modified following comments from the second reviewer and then some development of the method. The revision of the methods showed for example that the use of concave hull is not necessary. The simplified methods is robust enough and it requires much shorter computational time.*

2. Power law:

Reviewer writes:
I understand why the power law was employed for accuracy/quality assessment but it is not the easiest section to comprehend. Perhaps it is my ignorance. Nevertheless, I think this section needs to be revised as casual readers will struggle – I did. I see is no reason why a simple buoy to SAR deformation comparison cannot be performed. The buoy data is available from the lead author (Itkin et al., 2015). Further, the two-way comparison is far more useful anyways and what casual readers will be looking for. I think the power law quality check metrics can still be included but the author needs to add some additional "bread and butter" comparison statistics for casual readers.

*I disagree with the reviewer on this point. Power law is 'the bread and butter' of the sea ice deformation comparisons. The scale differences (in space and time) between the datasets do not allow for any other comparison (and for ship radar they were not even stored). This part of the paper is crucial in the way that it justifies the realism of the damage parcel method. The third*

*panel, panel c in Figure 6 was created explicitly with the though of the readers that are not very familiar with the strain rates. The third paragraph gives some practical use of the power law data interpretation:* *'A practical use of recalculating the strain rates to total displacement is also in observation or simulation design. To resolve, for example, 100 m displacements (small leads), 10 m displacements (large ridges), or 1 m displacements (cracks), accurate measurements at 200 m, 30 m, or 1 m resolution at daily scale are required.'*

*I believe it is good that this analysis is in a separate chapter that those who want can look at it in detail, while the rest of the readers can focus on the other parts of the paper.*

*I have reviewed the text and improved some minor inconsistencies. The caption of Figure 6 was extended now to:*
*'Power law for the spatial scales of ship radar-, buoy- and SAR-derived sea ice deformation calculations: a) original values from ship radar (purple shades with means for logarithmically distributed length classes as orange rectangles) and buoys (green shades with means as circles), SAR-derived deformation (blue shades with stars as means) and DL filtered ship radar and buoy values, and c) total displacements. Total displacements in panel c have a matching y-axis, but the x-axis to stretched to the left to show extrapolation of the power law to 1-m scale displacements. The power laws are fitted to the length-scale classes mean. Power laws with long dash lines include all data and the short dash lines power laws only include the DL filtered data. DL and max values are thin black dash lines on all plots.'*

3. Lack of synoptic data:

The reviewer writes:
On Line 10 you state, "Our results revealed a cyclically changing winter sea ice cover, marked by synoptic events and transitions from pack ice to the marginal ice zone." However, this really was not investigated in the paper. There is no synoptic data in the paper. Again, casual statements like these will leave readers confused because this analysis is nowhere to be found in the paper. Why not add some supporting synoptic data (spatially) to make the manuscript more comprehensive?

*The weather time series from Graham et al, 2018 (Scientific Reports) are available and could be used for interpretation of the N-ICE2015 data. They are the base of the shading on Figure 9. I have considered adding these synoptic data to Figure 10, but I came to conclusion that this would sidetrack the manuscript from the main purpose of the method description. I have decided to remove (or soften greatly) all references to synoptic events from this paper. No such analysis is now highlighted in the abstract or conclusions. For clarity however, Figure 10 now includes labeling of the storm events (grey shading) and the text about the storms is better referenced to this figure.*

*Instead, I added an outlook statement in the conclusions. The MOSAiC data in the inner pack ice with longer time series offer a better opportunity for synoptic event analysis and the work is already underway:*

*'In the follow-up paper the method will be employed to the longer time series over the entire winter season (e.g. MOSAiC). This will allow for the analysis of the processes such as 'reactivation' of damage parcels after a long 'relaxation time' and seasonal development of CDCs.'*

4. Too many acronyms:

The reviewer writes:
There are so many acronyms and notation that the reader often forgets or has to refer back to what the definition is. There is nothing wrong with spelling things out in full and in fact it makes your paper more accessible to casual readers. Considering removing some of the notation for text.

*I agree with the reviewer that lots of acronyms makes a paper hard to read. In this manuscript I introduced 3 new (non-standard) acronyms: DL, LKFF and CDC. In addition there is a longer list of*

*the old established ones: LKF, SAR, GRD, GPS, MIZ. One project name: N-ICE2015. And a number of established mathematical symbols: sigma_x, lambda, A (area), epsilon_tot, alpha, beta.*

*For this revision I removed the DP abbreviation for 'damage parcels'. From the more standard ones I also don't use ESA, FYI (first-year ice), SYI (second-year ice) and EW (Extra Wide) acronyms anymore.*

Here the minor comments are addressed one-by-one:

Line 19: What implications? A good to idea to state what they are i.e. For example, …
More text has been added in this point of the introduction: 'For example, occurrence of leads locally increases the thermodynamic coupling between the atmosphere and ocean, increases light availability for the primary production and aids the navigation. On contrary, does the occurrence of pressure ridges increase the surface roughness and dynamical coupling between the atmosphere and ocean, provide protective habitat for the life in the ocean, and obstruct the navigation. '

Lines 22-25: How can increased deformation erode the long-term memory of ice thickness?  As I read Mitch's paper he and co-authors state predictability is lost with the onset of melt. Or are you suggesting winter-time deformation will complicate winter ice thickness retrievals? You need to be explicit about the link between deformation and seasonal prediction.

This is truly not specific enough and it was removed from the revised text. To balance the addition of the text added for the previous comment, I simplified this part of the introduction to:

'Deformation changes sea ice thickness instantly \citep{kwok2015,itkin2018,albedyll2022},  and gradually, though preferential melt rates of pressure ridges \citep[e.g.][]{salganik2023b} , both influencing the state of sea ice, critical for accurate sea ice forecasting and projections \citep{bushuk2017,tian2021}.'

Line 49: Those are not really references related to RADARSAT-1 and RADARSAT-2.  I suggest the following:

Mahmood, A., Crawford, J.P., Michaud, R., and Jezek, K.C. 1998. "Mapping the world with remote sensing." Eos, Transactions, American Geophysical Union, Vol. 79(No. 2): pp. 17, 23

Z. Ali, I. Barnard, P. Fox, P. Duggan, R. Gray, Peter Allan, Andre Brand & R. Ste-Mari (2004) Description of RADARSAT-2 synthetic aperture radar design, Canadian Journal of Remote Sensing, 30:3, 246-257, DOI: 10.5589/m03-078

I agree, those references are not appropriately placed. The text is now rewritten to:

'including data from RADARSAT program of the Canadian Space Agency and Sentinel-1 mission of the European Space Agency.'

Further down the text (data description) I removed mentioning ESA again.

Line 53:  I think the RGPS has some done a lot more than derive scaling laws and intersection angles with respect to understanding sea ice dynamics.

To keep it brief, I generalized this sentence to 'instrumental in deriving properties such as scaling laws in spatio-temporal distribution \citep{marsan2004,bouillon2015,rampal2019} and patterns in LKFs \citep{hutter2022}. '

Line 54-55: The spatial resolution of "deformation estimates from SAR" has been…

Text added.

Line 62-65; Redundant.  You just stated most of this information in the previous paragraph.

Also in accordance of definition of CDE (major comment 1), these two paragraphs have been merged into:

'This paper utilized sea ice deformation data and findings from the N-ICE2015 expedition conducted in January and February 2015 in the pack ice north of Svalbard, as reported by \cite{granskog2018}, \cite{itkin2017}, \cite{oikkonen2017} and \cite{graham2019}. The methodology involved comparing the power law of SAR-derived strain rates with other N-ICE2015 data. In the next phase, this study examined if SAR data can be employed to track damage along LKFs between temporally separate weather events. Finally, the paper explored the possibility of detecting CDCs.'

Line 70: You already defined SAR.

Removed.

Line 78: As with previous comment

Redefinition of CDC abbreviation removed.

Line 99: How where the SAR images pre-processed? Were they calibrated? I think some details on this is required.

The SAR images were not preprocessed and used uncalibrated. This this more clear I write:

'While not calibrated, each image was multi-looked, averaging radar intensity values over an 80 m by 80 m area.'

Line 378: The Conclusions do not really match (are missing) some of the items presented in the Introduction.

I believe this comment is mainly connected to the statement of the winter ice pack characteristics. These claims were removed from the abstract and conclusions as this paper focuses on the method and the follow-up paper will use the more extensive MOSAiC dataset to provide the analysis of deformation in the light of synoptic events.

Line 400: Can something be said as to the applicability of these techniques to summertime conditions? Or are these strictly limited to the winter time?

In brief: the LKFs do not exist in the summer, the CDCs break into individual ice floes (not clusters anymore), SAR looses brightness contrast, displacements are faster and very hard (often impossible) to detect. Some parts of the presented methods could be adopted, but the presented method is likely not applicable to the summer sea ice. In the text this is touched indirectly by saying:

'The method was also successfully applied in the marginal ice zone (MIZ), albeit the method fails in the areas directly at the ice edge and open water areas.'

My preference at this point would be not to mention summer at all as it would require appropriate discussion of considerable length.

**Final response to reviewer Andy Mahoney**

I thank Andy for his investment in the review, a very positive review and a large number of valuable suggestions!

Here are the major comments addressed one-by-one:

1. Methods section would benefit from restructuring

Reviewer writes:
I feel that readability of the methods sections could be improved with some restructuring of the text. Specifically, I would encourage the author to adopt a more chronological approach with description of the different steps involved in deriving the various results. For example, I find it highly non-intuitive that the text in section 3 goes into significant detail regarding the derivation of deformation components from triangulated ice motion data before the method for determining ice motion has been described in section 3.1.
On line 87, the text states "information from individual images was processed into displacements, drift, and deformation" and I think this would be a more appropriate order in which to describe the methodology to the reader (i.e., displacement and drift before deformation). However, at the same time, I also recommend reviewing the manuscript for consistent use of these terms. In particular, I cannot determine the difference between "drift" and "displacement" and the terms appear to be used interchangeably later in the text, which makes it a little confusing to list them separately here. I recommend choosing one term and using it consistently.

*The intention of this unusual structure was to emphasize that the buoy and ship radar data were processed in previous work, but that the basic method is identical. As this only confused both reviewers, I restructured it more logically. Now the 'Data' description of the SAR is slightly shortened. The SAR displacement calculation is described in the beginning of the Method section, followed by the basic sea ice deformation calculation method description. The additional SAR deformation processing section was renamed to 'Additional SAR sea ice deformation processing'.*

*I have also merged the old Figures 2 and 3 into one and improved the flow chart.*

*The use of drift and displacement has been carefully reviewed through the manuscript.*

2. More detail needed regarding definition of damage parcels

Reviewer writes:
I think I understand how the damage parcels are defined, but I had to read section 3.3 multiple times to do so. For me, it would help to explicitly clarify the relationship between the location and distribution of damage parcels and the nodes of the triangulated grid. I'm a visual thinker and I believe many other readers are too, so I would encourage the use of an additional figure to help crystalize this important concept.

*I have added an explanatory text:*
*'At the beginning of the damage-tracking procedure the entire area is seeded by equally-spaced undamaged parcels. In subsequent steps each parcel is tracked and its value of damage, convergence or divergence accumulates by the deformation value found within the search radius. At the same time its location is updated. This procedure is repeated for all time steps. After the last time step the accumulated values of damage are used to classify each parcel into predominately leads, ridged ice, mixed class and undamaged ice. Figure \ref{figX} give an overview of the damage parcel tracking method. '*

*and a figure where I explain the sequence of steps for the deformation is tracking.*

3. Clearer definitions of spatial scales

Reviewer writes:
On line 144, the text equates a value of "$\lambda \gg 679\ m$" with "scales approximately 10 km" (sic). I feel more explanation is needed here to explain how "scale" is defined here and how it is quantitatively related to the value of.

*The statement is now simplified to:*

'this condition is satisfied only when $\lambda >> $ 679 m, for example at an order of magnitude larger values of 5-10 km.'

4. Definition and use of CDE shape parameters could be improved

Reviewer writes:
I have a number of comments about the definitions and explanations of the CDE shape parameters used in this manuscript. These are detailed in the sub-comments below, but in short, I recommend that the author considers alternative, potentially more suitable statistics and define them in more detail in the text. In particular, I encourage the use of a figure to help explain some of the details.

- 4a Shape diameters and radii: The bulleted text in section 3.4 makes a number of references to the diameter or radius of CDEs, but does not provide the reader with the specific details necessary to understand how these terms are applied to highly non-circular shapes like those of many CDEs. From context, I expect the "max and min diameter" used in the definition of roundness (line 236) is measured using either caliper distances or the dimensions of an enclosing rectangle, though in either case these are not strictly diameters since they do not necessarily pass through the center of the shape. Similarly, the "mean diameter" used in the definition of fragmentation (line 233) is not sufficiently defined. It could be the average of the minimum and maximum diameters, though a more common definition is the diameter of a circle with equal area to the shape in question. There are also other definitions of diameter and radius that would require explicitly defining the center of the CDE.

- 4b Fragmentation parameter: I had difficulty understanding how the definition of this parameter on lines 233-236 could be related to fragmentation of CDEs until I looked ahead to Figure 9 (hence why I recommend using an additional figure at this point in the manuscript). As defined, it is a measure of the complexity or non-circularity of the CDE perimeter, which does not necessarily have anything to do with fragmentation. An alternative statistic that might be more sensitive to how far the perimeter penetrates into the interior of the CDEs is the convexity, which can be defined as: *concavity = convex hull perimeter / CDE perimeter*. This will take a value of 1 if the shape is convex or less than one if the perimeter has concavities. One advantage of this metric is that there is no underlying reliance on the properties of a circle. Also, there is a possible variant of this approach whereby the perimeters of all LKFs within convex hull (or just their skeletonized lengths) are also included with the CDE perimeter in the denominator. This would take account of all the unconnected LKFs that are currently encapsulated within CDEs without contributing to the fragmentation statistic.

- 4c Roundness parameter: As defined, this parameter could just as well be called "squareness" (a square would give the same value as a circle), but the parameter is really a measure of aspect ratio or elongation. A more suitable measure of roundness or circularity could be the ratio: 4 Pi$\pi$ · *CDE area* / (*convex hull perimeter*)**2. By using the convex hull perimeter (instead of the CDE perimeter), this metric is sensitive to overall elongation and angularity of the shape while being insensitive to the properties that are indicative of fragmentation.

- 4d Distance between LKFs: The distance between LKFs is defined as "min and max CDE radius" (line 239), but I suspect this should read "diameter" instead of radius. Please also see comment 4a above regarding detailed definitions of such dimensions.

*I really appreciate the input the reviewer made to this point! These are all excellent comments that were taken into account and by doing this greatly improved the paper. The updated term is now 'Coherent Dynamic Clusters' (CDC) and it is better defined in the introduction. In the method section CDCs are now described as:*
'As defined in the introduction  CDCs are clusters of ice plates, containing damaged or undamaged ice, that at a given moment of time move coherently within each other and differentially in respect to the other CDC that are disconnected from them with fractures like LKFs. CDCs are transient features that only describe the state of ice pack in a certain moment of time. A statistical description of geometrical characteristics of CDCs and LKFs can therefore be used to evaluate the state of ice over time. Such statistical description suggested here was determined by using the

*location information of the deformation calculations. The triangle nodes filtered by LKFF were used to calculate polygons in computational geometry operations such as polygon union, distance buffering, difference (removed intersection) and removal of polygon holes as shown on Figure \ ref{fig4}. The resulting polygons were classified into CDCs and LKFs and the following characteristics were determined at each time step:*

- *\textit{CDC size} and \textit{CDC density} are measured as area of the CDC ($km^{2}$) and number of CDC per total area $A$ ($km^{-2}$). At time steps with very low sea ice deformation the entire region may be divide into just one or a few large CDC, while at time steps with strong deformation it may divide into many small CDCs. CDC size is a variable more appropriate for mapping, while time series regional comparisons may be better achieved by comparing CDC density.*
- *\textit{CDC circularity} is a measure of roundness and compactness of CDCs. It is determined by isoperimetric quotient - the ratio of CDC area $A$ and that of the circle having the same perimeter: $\frac{4 \pi A}{p^2}$, where $p$ is the convex hull perimeter of CDC polygon. Completely circular CDC scores value 1, while smaller values are typical for less compact shapes (e.g. square score corresponds to 0.75). Convex hull perimeter is used instead of the CDE perimeter due to its sensitivity to overall elongation and angularity of the shape while being insensitive to the properties that are indicative of fragmentation. To compare the CDC shapes to the summer floe shapes values \ citep[e.g.][]{hwang2022}, more frequently used \textit{CDC roundness} was calculated. Roundness is less sensitive to 'squareness' as it is a simple ratio between max and min diameter of CDC (see next point), with values close to 1 for circular CDE.*
- *\textit{CDC complexity} is a measure of fragmentation and it is determined by a ratio between CDC perimeter and its mean diameter. The mean diameter is the average of the min and max diameter estimated from min and max radius. First the centroid of the CDC is calculated. The min and max radii are then the shortest and the longest distance to the CDC boundary. CDC complexity has a theoretical value close to 2 for simple elongated polygons (practically a very slim rectangle). Typical values span from 4 for parallelograms expected for sea ice, and 7 for complex polygons with meandering boundaries. The latter is typical for CDCs where the LKFF likely fails to detect small deformation and correctly divide such into several smaller CDCs. Large CDC complexity is therefore a measure of error of the method.*
- *\textit{LKF fraction} is LKF area per total area. It is determined in two separate methods: 1) by summing the area of of LKFF filtered triangles (min LKF fraction), and 2) by summing the area not covered by CDCs (max LKF fraction). The first method is more conservative than the latter, while the latter gives another measure of undetected deforming ice parcels.*
- *\textit{Distance between LKFs} is estimated from the min and max CDC diameter used to estimate CDC complexity (see above). Based on the diameters min and max distance are estimated.*

*I have also improved the flowchart of the method with examples and a drawing.*

5. Figure sub-panels

Reviewer writes:
This is a minor comment, but applies it applies to every multi-panel figure. I encourage the author to label each sub-panel with a letter and use this an identifier in the text instead of relative descriptors like "left" and "right". I understand this is partly stylistic, but I feel it removes any ambiguity when cross-referencing sub-panels both in the caption and in the main text (e.g., lines 341 and 346). It may also be required by the journal.

The figure sub-panels were where necessary marked by alphabetic indexes.

Here the minor comments are addressed one-by-one:

Line 144: I think there's an "of" missing before "approximately".

Added.

Line 153: I recommend adding letters to identify each panel (see comment 5 above). Although the references to "left", "right", etc in the caption are technically accurate, I find it would be helpful to explicitly identify that each instance references more than one panel in the figure.
Done.

Line 162: Replace "of" with "by"
Done.

Lines 167 & 168: I assume the author intended to remove these question marks before submission. Do they signify missing citations?
Yes, citation was missing – fixed now.

Line 170: What is "DLDL"? Is this simply a typo for DL?
Typo fixed.

Lines 170-171: I feel a little more explanation of Bouillon and Rampal's smoothing technique is necessary here. I do not think it should be necessary to be familiar with a cited work to understand the text in which the citation is made. Also, in this case, the text refers to a "kernel size of 3 triangles", whereas Bouillon and Rampal define the size of their kernel in terms of the number of vertices.

Text expanded to:
'While this problem can be mitigated by isotropic smoothing \citep{lindsay2003}, \cite{bouillon2015} suggested a directional filtering of deformation values of triangles specifically along the LKFs. Such anisotropic smoother, here called the LKF filter (LKFF) follows the direction of the LKF and preserves the accurate information of the deformation localization. LKFF was defined by the size of the kernel, the number of boundary crossed in each direction, and its minimal size. LKFF was applied to the data previously filtered by DL.The kernel size suggested by \citep{bouillon2015} was 3, for a dataset with 10-km grid spacing. In this study, the spacing is much shorter (800 m) and the smallest possible kernel of 1 boundary crossing was used. For a single LKF this resulted in LKFF size of 3 triangles, while for a complex case of LKF crossing the kernel size may be larger. '

Figure 4: In its current form I do not feel this figure gives the reader much more information than is already provided in the text. Elsewhere, I recommend the inclusion of additional figures (see comments 2 and 4 above), so if space is an issue this figure could possibly be omitted. However, if the authors choose to keep the figure, then I feel some explanation should be given for the different colors used for boxes and arrows.

This Figure is now merged (co-located with a sample figure). The colors and arrows are also better explained. I feel this will be a nice figure to use in presentations to audience.

Line 209: Insert "each" between "in" and "pair"
Done.

Line 213: There are some unnecessary parentheses in this citation
Removed.

Lines 214-215: See comment 1. Are displacements and sea ice drift the same thing?
Now reviewed in the entire text. We basically only operate with displacements in this paper (although mean drift velocities are used in deformation calculations).

Line 223: Replace "nods" with "nodes"
Done.

Line 233: If the author chooses to keep their definition of fragmentation (see comment 4b above) then I would replace "circumference" on line 233 with "perimeter", since the term circumference is only defined for a circle. Also, according to the definition given, a perfect circle would have a fragmentation value of 2π, not 0.
This have been reviewed and changed.

Figure 5: Similar to my comment for Figure 4, I do not feel figure 5 adds much value to the text. Instead, I think I would find it more useful to see a graphical representation of how nodes associated
with LKFs and CDEs are enclosed in convex hulls (see comment 4 above).

This figure now got additional examples and an illustration.

Figure 6: The axis labels are quite difficult to read and the neither the caption nor the legend explain the meaning of the different colors. Also, I recommend using the same units and limits for the x- axes of all three plots.
I have labeled the panels with a,b,c and extended the caption to:
'Power law for the spatial scales of ship radar-, buoy- and SAR-derived sea ice deformation calculations: a) original values from ship radar (purple shades with means for logarithmically distributed length classes as orange rectangles) and buoys (green shades with means as circles), SAR-derived deformation (blue shades with stars as means) and DL filtered ship radar and buoy values, and c) total displacements. Total displacements in panel c have a matching y-axis, but the x-axis to stretched to the left to show extrapolation of the power law to 1-m scale displacements. The power laws are fitted to the length-scale classes mean. Power laws with long dash lines include all data and the short dash lines power laws only include the DL filtered data. DL and max values are thin black dash lines on all plots.'

Line 248: I think "similar" may be a more suitable word than "resembling"
Changed.

Line 249: Replace "ad" with "and"
Done.

Lines 252-253: The text "This increased $\alpha$ and $\beta$ for ship radar to 24.61 and -0.7, respectively" is confusing to me. The value of is greater than that reported earlier in the paragraph for the full suite of data, but value of β represents a decrease. Some rewording may be necessary here.
This was revised in the entire paragraph.

Lines 256-265: This paragraph appears to be a duplicate of the preceding paragraph.
Paragraph was merged and redundant text removed.

Line 266: Replace "amount" with "number", since this refers to a countable quantity.
Done.

Lines 267-268: This could be re-written as "... increased $\alpha$ from 4.79 to 14.63 and $\beta$ from -0.14 to -0.59" to both improve clarity of the text and remove the need to use "respectively". This practice could be adopted in other place in the text too.
This was revised in the entire paragraph.

Line 277: Specify units after "100" (presumably meters).
Unit added.

Line 278: I'm not sure what "boarding" means in this context. Is this a typo? I would recommend using either "opening" or "widening" instead.
Typo corrected.

Figure 8: I think I would understand and appreciate this figure more fully if I was confident I understood how the damage parcels were defined and located. Please refer to my comment 2 above.
This should now be better understandable with added text, schematic annotation on this figure.

Line 325: I assume this question mark was not intended to be included in the text, like those on lines 167and 168.
Missing citation added.

Line 326: I do not think Murzda et al's paper supports the assertion that "large fractures ... healed slowly". First, Murzda et al report crack healing on timescales of "tens to hundreds of seconds", which I would characterize as quite rapid in this context. Second, I'm not convinced their lab-based observations can easily be scaled up to that of the "large fractures" described in this study. Murzda et al explicitly note this at the end of section 1 of their paper.
This is a very interesting topic that I would like to study further, but a good analysis deserves a separate paper. The sentence has been rephrased into:
'This finding aligns with previous studies and indicates that large fractures, where shear and shape mismatch occurred, healed slowly \citep{coon2007, oikkonen2017}.'

Line 335: The "continuous curve with increased parcel density" is a pretty subtle feature. I recommend labeling it on this figure with some form of annotation.
Annotated.

Figure 9: The significance of the gray regions in the time series plots is not explained in the caption. From indirect cues in the main text, I assume they indicate the occurrences of storms. In addition to explaining their meaning in the caption, I also encourage the author consider naming and labeling each storm uniquely (e.g., "mid-January storm", etc, or more simply Storm 1, ...). This would allow easy and clear cross-referencing in the main text. I also encourage labeling each sub-panel with a letter, as per comment 5 above.
The figure itself and the caption were improved as recommended.

---

## Referee Report (RR1)

Novel methods to study sea ice deformation, Linear Kinematic Features and Coherent Dynamic Clusters from imaging remote sensing data by Polona Itkin

General comments
This revised version of the paper is much easier to follow and the novelty of the CDC's is evident. I appreciate the author taking the time to explain things more clearly. I still think the validation/comparison could be simplified more for casual readers but perhaps that is just my unfamiliarity with that approach. I did notice many type-o's throughout that require attention. I noted several of them but a more thorough proof-read is required to catch them all – this is very minor. Overall, this is a strong contribution to the sea ice dynamics field.

Stephen Howell, ECCC

Specific Comments (mostly type-o's and I am sure there are more).
Line 20
Remove "does"
change increase to increases
provide to "providing a"
obstruct to obstructing

Line 33
Replace deep with better

Line 149
Replace Figure ?? with the correction Figure reference.

Line 156
LKF has already been defined.

Line 169
LKF has already been defined.

Lines 196-197
Move "were" to after "study"

Editor
This paper has some good elements.  The CDE framework is novel but poorly defined and this needs to be revised. The power law is useful but I found it a challenge to follow.  I see no reason why simple buoy comparison statistics cannot be used as well.  I think readers will struggle with many sections of these paper in terms of readability. Overall, this paper has new elements but the presentation needs to be improved.

Novel methods to study sea ice deformation, linear kinematic features and coherent dynamic elements from imaging remote sensing data by Polona Itkin

Summary
Automatically identifying sea ice dynamic features is challenging. In this paper the author presents several new methods to estimate several dynamic features from SAR imagery using the N-ICE2015 study period as a test case. I like the ideas and methods presented in this paper and they certainly add the understanding of sea ice dynamics. This paper has new elements however, I found some sections and items challenging to fully grasp. I think this paper can be published it just requires some revisions to improve its presentation (readability, clarity, etc.). I hope my comments help the author improve this work.

General Comments (major)
1. I really like the idea of CDE's but their definition is a bit confusing.  It seems to me CDE's are an architecture or framework or terms (not term singular) that certain variables can be used to collectively describe winter pack ice. Am I right? However, you first define Coherent Dynamic Elements (CDE) as the boundary of rigid ice plates (Line 58 and 59). OK. In the Abstract you say CDE describes the behaviour of the winter pack but nothing in the paper including your Conclusion relates the winter pack behaviour during N-ICE2015 in that context. I thought I was missing something. Further, if a new term is introduced, then the definition must be consistent. Your definition and usage of CDE needs revision throughout the text otherwise readers will be scratching their heads as its meaning and usage. I suggest defining the CDE framework (with associated variables) earlier in the paper and explicitly describe how these terms can be used collectively to understand winter pack behaviour with evidence from N-ICE2015.

2. I understand why the power law was employed for accuracy/quality assessment but it is not the easiest section to comprehend.  Perhaps it is my ignorance.  Nevertheless, I think this section needs to be revised as casual readers will struggle – I did.  I see is no reason why a simple buoy to SAR deformation comparison cannot be performed.  The buoy data is available from the lead

author (Itkin et al., 2015). Further, the two-way comparison is far more useful anyways and what casual readers will be look for. I think the power law quality check metrics can still be included but the author needs to add some additional "bread and butter" comparison statistics for casual readers.

3. On Line 10 you state, "Our results revealed a cyclically changing winter sea ice cover, marked by synoptic events and transitions from pack ice to the marginal ice zone." However, this really was not investigated in the paper. There is no synoptic data in the paper. Again, casual statements like these will leave readers confused because this analysis is nowhere to be found in the paper. Why not add some supporting synoptic data (spatially) to make the manuscript more comprehensive?

4. There are so many acronyms and notation that the reader often forgets or has to refer back to what the definition is. There is nothing wrong with spelling things out in full and in fact it makes your paper more accessible to casual readers. Considering removing some of the notation for text.

Specific Comments (minor)
Line 19
What implications? A good to idea to state what they are i.e. For example, …

Lines 22-25
How can increased deformation erode the long-term memory of ice thickness? As I read Mitch's paper he and co-authors state predictability is lost with the onset of melt. Or are you suggesting winter-time deformation will complicate winter ice thickness retrievals? You need to be explicit about the link between deformation and seasonal prediction.

Line 49
Those are not really references related to RADARSAT-1 and RADARSAT-2. I suggest the following:

Mahmood, A., Crawford, J.P., Michaud, R., and Jezek, K.C. 1998. "Mapping the world with remote sensing." Eos, Transactions, American Geophysical Union, Vol. 79(No. 2): pp. 17, 23

Z. Ali, I. Barnard, P. Fox, P. Duggan, R. Gray, Peter Allan, Andre Brand & R. Ste-Mari (2004) Description of RADARSAT-2 synthetic aperture radar design, Canadian Journal of Remote Sensing, 30:3, 246-257, DOI: 10.5589/m03-078

Line 53
I think the RGPS has some done a lot more than derive scaling laws and intersection angles with respect to understanding sea ice dynamics.

Line 54-55
The spatial resolution of "deformation estimates from SAR" has been…

Line 62-65

Redundant. You just stated most of this information in the previous paragraph.

Line 70
You already defined SAR.

Line 78
As with previous comment

Line 99
How where the SAR images pre-processed? Were they calibrated? I think some details on this is required.
Line 189
See General Comment #1.

Line 378
The Conclusions do not really match (are missing) some of the items presented in the Introduction.

Line 400
Can something be said as to the applicability of these techniques to summertime conditions? Or are these strictly limited to the winter time?

Figure 2 and 3:
Probably a good idea to note in the Figure Caption the artifacts or bad data presented (Line 145)

---

## Author Response (AR2)

Dear Andy, Stephen and Vishnu,

Thank you an efficient review process! I have implemented all of your final comments and improved Figure 1. The method code is now published on Zenodo and the reference is added to the manuscript.

Kind regards,
Polona